

# Global evaluation of runoff from ten state-of-the-art hydrological models

Hylke E. Beck[1], Albert I.J.M. van Dijk[2], Ad de Roo[1], Emanuel Dutra[3], Gabriel Fink[4], Rene Orth[5], and Jaap Schellekens[6]

[1]European Commission, Joint Research Centre (JRC), Via Enrico Fermi 2749, 21027 Ispra (VA), Italy
[2]Fenner School of Environment & Society, Australian National University (ANU), Canberra, Australia
[3]European Centre for Medium-Range Weather Forecasts (ECMWF), Redding, UK
[4]Center for Environmental Systems Research (CESR), University of Kassel, Kassel, Germany
[5]Institute for Atmospheric and Climate Science, ETH Zurich, Switzerland
[6]Inland Water Systems Unit, Deltares, Delft, The Netherlands

*Correspondence to:* Hylke E. Beck (hylke.beck@gmail.com)

**Abstract.** Observed runoff data from 966 medium sized catchments (1000 to 5000 km$^2$) around the globe were used to comprehensively evaluate the daily runoff estimates (1979–2012) of six global hydrological models (GHMs) and four land surface models (LSMs) produced as part of Tier-1 of the eartH2Observe project. The models were all driven by the WATCH Forcing Data ERA-Interim (WFDEI) meteorological dataset, but used different datasets for non-meteorologic inputs and were run

at various spatial and temporal resolutions, although all data were re-sampled to a common $0.5°$ spatial and daily temporal resolution. For the evaluation, we used a broad range of performance metrics related to important aspects of the hydrograph. We found pronounced inter-model performance differences, underscoring the importance of hydrological model uncertainty in addition to climate input uncertainty, for example in studies assessing the hydrological impacts of climate change. The (uncalibrated) GHMs were found to perform, on average, better than the (uncalibrated) LSMs in snow-dominated regions, while

the ensemble mean was found to perform only slightly worse than the best (calibrated) model. The inclusion of less reliable models did not appreciably degrade the ensemble performance. Overall, more effort should be devoted on calibrating and regionalizing the parameters of macro-scale models. We further found that, despite adjustments using gauge observations, the WFDEI precipitation data still contain substantial biases that propagate into the simulated runoff. The early bias in the spring snowmelt peak exhibited by most models is probably primarily due to the widespread precipitation underestimation at high

northern latitudes.

## 1 Introduction

Hydrological models are indispensable tools for many purposes, including but not limited to, (i) flood and drought forecasting, (ii) water resources assessments, (iii) assessing the hydrological impacts of human activities, and (iv) increasing our understanding of the hydrological cycle. It is more than 50 years since the first attempts at hydrological modeling (Lins-

ley and Crawford, 1960; Rockwood, 1964; Sugawara, 1967; Freeze and Harlan, 1969). Since then, a plethora of conceptual, physically-based, and stochastic hydrological models has been developed, each with its own assumptions and characteristics





(for non-exhaustive overviews, see Singh, 1995; Singh and Frevert, 2002; Rosbjerg and Madsen, 2006; Trambauer et al., 2013; Sooda and Smakhtin, 2015; Bierkens et al., 2015; Kauffeldt et al., 2016). Because all hydrological models are inevitably imperfect representations of reality, they produce highly uncertain estimates even if we would have access to perfect meteorological data (Beven, 1989).

The quantification of these uncertainties using independent data sources is of critical importance to advance model development, reject deficient model structures and parameterizations, quantify model credibility, and ultimately bring some order in the plethora of models (Klemeš, 1986; Wagener, 2003; Döll et al., 2015; Clark et al., 2015). There have been several collaborative research efforts focusing on the intercomparison and verification of hydrological models. The earliest were coordinated by the World Meteorological Organization (WMO, 1975, 1986, 1992). Other noteworthy initiatives include the Model Parameter

Estimation Experiment (MOPEX; Duan et al., 2006), the Global Soil Wetness Project (GSWP; Dirmeyer, 2011), the Water Model Intercomparison Project (WaterMIP; Haddeland et al., 2011), and the Global Energy and Water Exchanges (GEWEX) LandFlux project (McCabe et al., 2016). These initiatives have led to numerous multi-model evaluation studies focusing on such hydrological variables as runoff (e.g., Gudmundsson et al., 2012; Zhou et al., 2012), evaporation (e.g., Schlosser and Gao, 2010; Jiménez et al., 2011; Miralles et al., 2015), soil moisture (e.g., Guo et al., 2007; Xia et al., 2014), snow cover (e.g., Slater

et al., 2001), and total water storage (Güntner, 2008), among others.

      One of the most useful variables for hydrological model evaluation is runoff, since it reflects the integrated response of a host of hydrological processes occurring in a catchment (Fekete et al., 2012) and because observations are readily available for many catchments across the globe (Hannah et al., 2011). Table 1 lists all macro-scale (i.e., continental to global scale) studies evaluating the runoff estimates of multiple models that have been published so far. Out of these 17 studies, two focused

on the conterminous USA, three focused on Europe, while twelve had a global scope. However, many of these studies used runoff observations from a relatively small number ($< 100$) of large catchments ($\gg 10\,000$ km$^2$). The use of a small number of basins limits confidence in the results and precludes a spatially detailed assessment, while the large size of the catchments makes it more difficult to distinguish between deficiencies in the forcing, the (sub-)surface component, or the river routing component of the modeling chain. Moreover, a large number of the studies only evaluated monthly mean runoff, neglecting the

important daily variability in runoff, or used the Nash and Sutcliffe (1970) efficiency (NSE), which is increasingly considered to be a flawed metric for model performance (Schaefli and Gupta, 2007; Jain and Sudheer, 2008). Furthermore, many studies considered only a few hydrological models ($\leq 5$) or performance metrics ($\leq 2$), limiting the insights that the evaluation might offer.

      As part of Tier-1 of the eartH2Observe project (http://www.earth2observe.eu), ten state-of-the-art hydrological models were

run globally at a daily time step for the period 1979–2012 using the same forcing dataset (Dutra, 2015). Six of the models are global hydrological models (GHMs) while four of the models are land surface models (LSMs). GHMs have traditionally been designed to simulate (sub-)surface water fluxes and storages, while LSMs have traditionally been designed to simulate the soil-vegetation-atmosphere interactions within climate models (Haddeland et al., 2011; Bierkens, 2015). GHMs generally represent hydrological processes in a more conceptual way, solve only the water balance, commonly operate at daily time steps,

and typically have a small number of soil layers ($\leq 3$ in the current study) and a single snow layer. Conversely, LSMs generally



**Table 1.** Overview of macro-scale (continental to global) studies evaluating the runoff estimates of multiple models, sorted by region and then publication date. The present study has been added for the sake of completeness.

| Study | Region | Number of models | Number of catchments (size range) | Evaluation time scale(s) |
|---|---|---|---|---|
| Lohmann et al. (2004) | Cont. USA | 4 | 1145 (23 to 10 000 km$^2$) | Daily, monthly, annual, long term |
| Xia et al. (2012) | Cont. USA | 4 | 969 (23 to 1 353 280 km$^2$) | Daily, weekly, monthly, annual, long term |
| Prudhomme et al. (2011) | Europe | 3 | 579 ($<$ 1000 km$^2$) | Daily |
| Gudmundsson et al. (2012) | Europe | 9 | 426 ($<$ 4000 km$^2$) | Daily, annual, long term |
| Greuell et al. (2015) | Europe | 5 | 46 (9948 to 658 340 km$^2$) | Daily, monthly, annual, long term |
| Milly et al. (2005) | Global | 12 | 165 ($>$ 50 000 km$^2$) | Long term |
| Decharme and Douville (2006) | Global | 6 | 80 (100 000 to 4 758 000 km$^2$) | Daily, monthly |
| Decharme and Douville (2007) | Global | 6 | 80 (100 000 to 4 758 000 km$^2$) | Monthly |
| Decharme (2007) | Global | 2 | 80 (100 000 to 4 758 000 km$^2$) | Monthly |
| Materia et al. (2010) | Global | 13 | 30 (82 000 to 4 677 000 km$^2$) | Monthly |
| Zaitchik et al. (2010) | Global | 4 | 66 (19 000 to 4 600 000 km$^2$) | Daily, annual |
| Haddeland et al. (2011) | Global | 11 | 8 (650 000 to 4 600 000 km$^2$) | Monthly |
| Zhou et al. (2012) | Global | 14 | 150 (not specified; $\gg$ 10 000 km$^2$) | Annual |
| Van Dijk et al. (2013b) | Global | 5 | 6192 (10 to 10 000 km$^2$) | Monthly |
| Beck et al. (2015) | Global | 4 | 4079 (10 to 10 000 km$^2$) | Daily, long term |
| Yang et al. (2015) | Global | 7 | 16 (135 757 to 3 475 000 km$^2$) | Monthly, annual |
| Zhang et al. (2016) | Global | 4 | 644 ($\gg$ 2000 km$^2$) | Monthly, annual |
| This study | Global | 10 | 966 (1000 to 5000 km$^2$) | Daily, 5-day, monthly, annual, long term |

represent hydrological processes in a more physically-based way, solve both the water and energy balances, typically operate at (sub-)hourly time steps, and tend to have many soil and snow layers (4–11 and 1–12, respectively, in the current study). The present study aims to comprehensively evaluate the runoff estimates of these ten models across the globe in an effort to answer the following pertinent research questions:

1. How well do the different models simulate runoff?

2. How well do the models perform in terms of long-term runoff trends?

3. How do the results of the GHMs differ, if at all, from those of the LSMs?

4. Are calibration and regionalization important or even essential?

5. What is the impact of the forcing data on the simulated runoff?

6. How valuable are multi-model ensembles for improving runoff estimates?

7. Do all models show the early bias in runoff timing in snow-dominated catchments previously documented (e.g., Zaitchik et al., 2010) and what is the cause?

We use daily runoff observations during 1979–2012 from a large, highly diverse, quality-controlled set of medium sized catchments. This leads to more reliable and generalizable conclusions, and allows us to explicitly compare the performance
15
among different climate types (Andréassian et al., 2007; Stahl et al., 2011; Gupta et al., 2014). Moreover, we use a broad range





of performance metrics, including runoff signatures (measures that quantify the hydrograph shape such as runoff coefficient and baseflow index; Olden and Poff, 2003; Monk et al., 2007) that can be related to specific hydrological processes (Yilmaz et al., 2008).

## 2  Data

### 2.1  Forcing

The models were all driven by the daily 0.5° WATCH Forcing Data ERA-Interim (WFDEI) meteorological dataset (1979–2012; Weedon et al., 2014) with the precipitation ($P$) data adjusted using the monthly 0.5° gauge-based Climate Research Unit (CRU) TS3.1 dataset (Harris et al., 2013). Although the models all used the same $P$ data, they used potential evaporation (PET) derived using diverse formulations, ranging from the temperature-based Hamon equation (PCR-GLOBWB) to various radiation-based approaches (WaterGAP3, SWBM, and HBV-SIMREG), the combination Penman-Monteith equation (HTESSEL, JULES, LISFLOOD, SURFEX, and W3RA), and a surface-energy balance approach (ORCHIDEE). The models also used different datasets for non-meteorologic inputs. For more details, see Dutra (2015).

### 2.2  Simulated runoff

Table 2 lists the ten state-of-the-art macro-scale hydrological models of which we evaluated the simulated daily (non-routed) runoff (mm d$^{-1}$). The data used in this study have been named Tier-1 and represent an initial run by all participating modeling groups (Dutra, 2015). All data were acquired through the eartH2Observe Water Cycle Integrator (WCI; http://wci.earth2observe.eu). Six of the models are GHMs (LISFLOOD, PCR-GLOBWB, SWBM, W3RA, WaterGAP3, and HBV-SIMREG) and four are LSMs (HTESSEL, JULES, ORCHIDEE, and SURFEX). The GHMs were all run at daily time steps and the LSMs at hourly and 15-minute time steps. The models were run at a 0.5° spatial resolution, with the exception of LISFLOOD and WaterGAP3, which were run at 0.1° and 0.08°, respectively. For the analysis, however, all model output was resampled to a common 0.5° spatial and daily temporal resolution. Four of the models were subjected to varying degrees of calibration to improve their parameters (LISFLOOD, SWBM, WaterGAP3, and HBV-SIMREG; see Section 4.4 for specifics). Details concerning the models can be found in Dutra (2015).

### 2.3  Observed runoff

Daily and monthly observed runoff data were used in this study to evaluate the runoff estimates of the models. The observed runoff and catchment boundary data used in this study originate from the same three sources as Beck et al. (2013, 2015, 2016a), namely (i) the Global Runoff Data Centre (GRDC; http://www.bafg.de/GRDC/), (ii) the Geospatial Attributes of Gages for Evaluating Streamflow (GAGES)-II database (Falcone et al., 2010), and (iii) an Australian runoff data compilation by Peel et al. (2000). The following seven criteria were used to select suitable catchments for our analysis:



**Table 2.** Overview of the hydrological models considered in this study. For definitions of the model name acronyms, see Dutra (2015). Definitions of model-class acronyms: GHM, global hydrological model; and LSM, land surface model.

| Model name | Data provider(s) | Reference(s) | Model class |
|---|---|---|---|
| HTESSEL | European Centre for Medium-Range Weather Forecasts (ECMWF) | Balsamo et al. (2009, 2011) | LSM |
| JULES | Natural Environment Research Council (NERC) | Best et al. (2011) | LSM |
| LISFLOOD | Joint Research Centre (JRC) | Burek et al. (2013) | GHM |
| ORCHIDEE | Centre National de la Recherche Scientifique (CNRS) | Krinner et al. (2005) | LSM |
| PCR-GLOBWB | University of Utrecht | Van Beek and Bierkens (2009) | GHM |
| SURFEX | Météo France | Decharme et al. (2011, 2013) | LSM |
| SWBM | Eidgenössische Technische Hochschule (ETH) Zürich | Orth and Seneviratne (2015) | GHM |
| W3RA | Australian National University (ANU) and Commonwealth Scientific and Industrial Research Organisation (CSIRO) | Van Dijk (2010) | GHM |
| WaterGAP3 | University of Kassel | Verzano (2009) | GHM |
| HBV-SIMREG | JRC | Beck et al. (2016a) | GHM |

1. The runoff record length was required to be $\geq 5$ years (not necessarily consecutive) during 1979–2012 (the temporal span of the simulated runoff data).

2. The catchment area had to be $< 5000$ km$^2$, to minimize the effects of channel routing delays and reduce the likelihood of significant anthropogenic water use. We could not use larger catchments and evaluate routed runoff estimates since three of the models did not not simulate river routing (JULES, SWBM, and HBV-SIMREG).

3. The catchment area had to be $> 1000$ km$^2$, to prevent catchments unrepresentative of the 0.5° grid cells (2182 km$^2$ at 45°N/S) from confounding the results.

4. To reduce human influences, catchments were required to have $< 2$ % classified as urban (using the "artificial areas" class of the GlobCover version 2.3 map; 300-m resolution; Bontemps et al., 2011) and subject to irrigation (using version 5 of the Global Map of Irrigation Areas—GMIA; 5-min resolution; Siebert et al., 2005).

5. We used the Global Reservoir and Dam (GRanD) database (v1.1; Lehner et al., 2011) to exclude catchments influenced by major reservoirs (defined by total reservoir capacity $> 10$ % of the mean annual runoff).

6. Catchments with forest gain or loss $> 20$ % of the catchment area (the threshold at which changes in runoff can generally be detected; Bosch and Hewlett, 1982) were excluded using version 1.1 of the Landsat-based forest change dataset (30-m resolution; Hansen et al., 2013).

7. To further reduce the number of disinformative catchments, all runoff records were visually screened for artifacts and anthropogenic influences (caused by, for example, diversions and impoundments). Furthermore, USA catchments flagged as "non-reference" in the GAGES-II database were discarded, and GRDC catchments for which the catchment boundaries could not be reliably determined were discarded (Lehner, 2012).





In total 966 catchments (median size 1970 km$^2$; median record length 19 y during 1979–2012) were found to be suitable for the analysis, of which 641 catchments have daily runoff data and 325 catchments (mainly located in Russia) have only monthly runoff data. The locations of the selected catchments will be shown in the Results section. All observed runoff data were converted to mm d$^{-1}$ using the provided catchment areas.

## 3 Methodology

### 3.1 Model evaluation

The simulated runoff of the models were evaluated in five ways. First, for each catchment, we calculated the differences $D$ $(-)$ between simulated and observed values of several runoff signatures. Table 3 lists the six runoff signatures selected including their computation from the period with simultaneous simulated and observed runoff. The baseflow index (BFI), square-root transformed 1st flow percentile (Q1), and square-root transformed 99th flow percentile (Q99) require daily (rather than monthly) flow data. To compute the flow timing (T50) from monthly data, we first computed daily time series from monthly time series using linear interpolation. The square-root transformed runoff coefficient (RC), square-root transformed mean annual flow (MAR), Q1, and Q99 values were square-root transformed to give more weight to small values. $D$ was computed according to:

$$D_q = \frac{Y_{q\,\mathrm{sim}} - Y_{q\,\mathrm{obs}}}{\sigma_q},\tag{1}$$

where $Y$ represent the values of the runoff signatures $(-)$, $\sigma$ the standard deviations of the transformed runoff signatures $(-)$, the $q$ subscript denotes the runoff signature, while the 'sim' and 'obs' subscripts refer to simulated and observed, respectively. The $\sigma$ values in Equation 1 represent the spatial variability in the runoff signatures across the landscape and are used to normalize the $D$ values. They were derived from the Global Streamflow Characteristics Dataset (GSCD) v1.9 (Beck et al., 2015; http://water.jrc.ec.europa.eu/GSCD; see Table 3) taking into account the entire ice-free land surface excluding deserts (defined by an aridity index $> 5$), with the exception of the T50 $\sigma$, which considers only the snow-dominated ice-free land surface. Next, the mean $D$ value over all catchments was computed (expressed by $\overline{D}$). $D$ and $\overline{D}$ values closer to zero correspond to better model performance.

Second, to evaluate the temporal variability of the simulated runoff time series, we computed Pearson linear correlation coefficients ($r$) between daily, log-transformed daily, 5-day, monthly, monthly climatic, and annual time series of simulated and observed runoff (termed $r_{\mathrm{dly}}$, $r_{\mathrm{dly\,log}}$, $r_{5\,\mathrm{day}}$, $r_{\mathrm{mon}}$, $r_{\mathrm{mon\,clim}}$, and $r_{\mathrm{yr}}$, respectively). The $r_{\mathrm{dly}}$, $r_{\mathrm{dly\,log}}$, and $r_{5\,\mathrm{day}}$ values were only computed for catchments with daily observed runoff data. If monthly data were not supplied by the data providers, monthly values were computed by simple averaging of the daily data only if $> 25$ non-missing values were available. Annual values were computed by simple averaging of the monthly data (either supplied or computed) only if $> 10$ non-missing values were available. We subsequently computed for each model and metric the mean $r$ value over all catchments, expressed by $\overline{r}$. The $r$ and $\overline{r}$ values range from $-1$ to $1$, with higher values corresponding to better model performance.



**Table 3.** The long-term runoff behavioral signatures considered for evaluating the model performance. The signatures were computed, for each catchment, from the entire record of simultaneous observed and simulated runoff. The $\sigma$ values represent the spatial variability in the runoff signatures across the landscape.

| Runoff signature | Units | Description | Evaluated flow aspect | Standard deviation ($\sigma$) |
|---|---|---|---|---|
| RC | – | Square-root transformed runoff coefficient, ratio of long-term runoff to $P$ | Water balance | 0.33 |
| MAR | $\sqrt{mm\ yr^{-1}}$ | Square-root transformed long-term mean annual runoff | Water balance | 11.21 |
| T50 | d | The day of the water year marking the timing of the center of mass of flow (Stewart et al., 2005). A water year is defined as the 12-month period from October to September in the Northern Hemisphere and April to March in the Southern Hemisphere | Seasonal flow distribution | 34.36 |
| BFI | – | Base flow index, the ratio of long-term baseflow to total runoff. The baseflow portion of the total runoff was computed following the procedure of Gustard et al. (1992), which takes the minima at five-day non-overlapping intervals and subsequently connects the valleys in this series of minima to generate baseflow. | Partitioning between quickflow and baseflow, flow peakiness | 0.18 |
| Q1 | $\sqrt{mm\ d^{-1}}$ | Square-root transformed 1st percentile exceedance flow | Peak-flow magnitude | 1.27 |
| Q99 | $\sqrt{mm\ d^{-1}}$ | Square-root transformed 99th percentile exceedance flow | Low-flow magnitude | 0.21 |

Third, to summarize the overall performance of each model, we computed for each catchment a summary performance statistic (termed OS) incorporating the previously mentioned metrics, and computed the mean value over all catchments ($\overline{OS}$). The OS consists of two parts, of which the first ($OS_{sig}$) considers the performance in terms of runoff signatures and is defined as:

$$OS_{sig} = 1 - \text{mean}\left[|D_{RC}|, |D_{MAR}|, |D_{T50}|, |D_{BFI}|, |D_{Q1}|, |D_{Q99}|\right]. \tag{2}$$

5  The second part ($OS_{var}$) evaluates the performance in terms of temporal variability, and is defined as:

$$OS_{var} = \text{mean}\left[r_{dly}, r_{dly\,log}, r_{5\,day}, r_{mon}, r_{mon\,clim}, r_{yr}\right]. \tag{3}$$

The summary score is subsequently computed following:

$$OS = \frac{OS_{sig} + OS_{var}}{2}. \tag{4}$$

The BFI, Q1, and Q99 components of Equation 2 and the $r_{dly}$ and $r_{dly\,log}$ components of Equation 3 were omitted if daily observed runoff data were unavailable for a particular catchment. Higher OS values correspond to better model performance; the maximum attainable value is 1.

10  Fourth, to evaluate the ability of each model to simulate the spatial variability in the six previously mentioned runoff signatures, Spearman rank correlation coefficients ($\rho$) were computed between simulated and observed values of the runoff signatures. Spearman rank correlation coefficients rather than Pearson linear correlation coefficients were used to minimize the influence of outliers. The $\rho$ values range from $-1$ to $1$, with higher values corresponding to better model performance.





Fifth, we computed trends in simulated and observed mean annual runoff time series (termed MAR trend) using the simple non-parametric approach of Sen (1968). We subsequently calculated the $\rho$ between simulated and observed MAR trends ($\rho_{\mathrm{MAR\,trend}}$), reflecting the agreement in spatial trend patterns.

Sixth and last, we produced for the four models for which PET data were available (ORCHIDEE, PCR-GLOBWB, W3RA, and WaterGAP3) density plots of grid cell values of aridity index (AI; ratio of long-term PET to $P$) versus runoff coefficient (RC; ratio of long-term runoff to $P$), revealing how the models behave in terms of RC under different climatic conditions.

For the evaluation, we used for each catchment the simulated runoff time series of the $0.5°$ grid cell with its center located within the catchment. However, if multiple grid cell centers were located within the catchment, we calculated the mean simulated runoff time series, and if no grid cell center was located within the catchment, we used the simulated runoff time series of the grid cell with its center located closest to the catchment centroid.

## 3.2  Multi-model ensembles

Ensemble modeling—using the outputs from multiple models or from different realizations of the same model—typically improves predictive accuracy and is widely used in atmospheric, climate, and hydrological sciences (Wandishin et al., 2001; Tebaldi and Knutti, 2007; Breuer et al., 2009; Viney et al., 2009). We tested two ways of combining the runoff estimates of the individual models into ensembles. First, for each $0.5°$ grid cell and day with non-missing values for all models, the mean simulated runoff of the ten models was calculated (i.e., equal weights were assigned to the models). The resulting runoff estimates will be referred to hereafter as "MEAN-All". Second, we computed the mean based on only the four models that performed best in terms of $\overline{\mathrm{OS}}$, to examine the effect of excluding less reliable models. These runoff estimates will be referred to hereafter as "MEAN-Best4".

## 3.3  Caveats

There are a number of caveats that should be kept in mind when interpreting the results. First, some of the models (notably the LSMs) were not traditionally developed to estimate daily runoff for such small catchments. Most of the GHMs, on the other hand, were specifically designed with runoff estimation in mind, and four were even explicitly calibrated against runoff observations (LISFLOOD, SWBM, WaterGAP3, and HBV-SIMREG; see Section 4.4 for specifics). Second, a model performing poorly in one respect may well perform better for other hydrological variables, climates, catchments, or performance metrics. Third, a poor model performance could simply be the result of suboptimal parameter values. Fourth, some studies have found that less reliable models may still lead to a better ensemble mean (Ajami et al., 2006; Viney et al., 2009), although this did not appear to be the case here (see Section 4.6). Fifth and finally, we stress that while some models may perform well, they are inherently unsuitable for specific types of impact assessments. For example, SWBM and HBV-SIMREG do not account for physical differences among land-cover types and hence cannot be used for studies assessing the hydrological impacts of changes in land cover.



## 4   Results and discussion

In this section we will answer the questions posed in the introduction.

### 4.1   How well do the different models simulate runoff?

Tables 4 and 5 show, for the uncalibrated models, the calibrated models, and the ensembles, (i) the mean difference between
simulated and observed values of the (normalized) runoff signatures ($\overline{D}$), (ii) the mean temporal correlation between simulated
and observed runoff time series ($\overline{r}$), and (iii) the mean overall performance in terms of runoff signatures and temporal correla-
tion coefficients ($\overline{OS}$). HTESSEL obtained negative $D$ values for the square-root transformed runoff coefficient (RC) and the
square-root transformed mean annual runoff (MAR), indicating it underestimates runoff. JULES performed relatively poorly
in terms of temporal correlation, as indicated by the low $r$ values. Conversely, LISFLOOD performed well overall, particularly
in terms of temporal correlation, although it tends to overestimate RC and MAR. ORCHIDEE appears to strongly underesti-
mate runoff and performed poorly in terms of temporal correlation, whereas PCR-GLOBWB shows moderate scores for all
metrics. Apart from a much too early bias in the flow timing (T50), SURFEX demonstrated fair performance overall. Similar
to SURFEX, W3RA exhibits a very early bias in T50, but generally obtained moderate scores. WaterGAP3 and particularly
HBV-SIMREG performed well for all metrics. JULES, ORCHIDEE, SURFEX, WaterGAP3, and especially SWBM displayed
negative $D$ values for the baseflow index (BFI) and the square-root transformed 99th flow percentile (Q99), and a positive $D$
value for the square-root transformed 1st flow percentile (Q1; Tables 4 and 5), suggesting they consistently overestimate quick-
flow. Conversely, LISFLOOD and particularly PCR-GLOBWB exhibited positive $D$ values for BFI and Q99, and a negative
$D$ value for Q1 (Tables 4 and 5), indicating they tend to underestimate quickflow.

Tables 4 and 5 also present, for the ten models and the ensembles, the spatial correlation between simulated and observed
values of the runoff signatures ($\rho$). HTESSEL, JULES, LISFLOOD, and W3RA obtained moderate to good performance for
all runoff signatures. ORCHIDEE performed poorly in terms of RC, MAR, T50, and Q1, while PCR-GLOBWB performed
poorly in terms of T50, BFI, and Q1. SURFEX showed particularly poor scores for BFI and the trend in mean annual runoff
(MAR trend), while SWBM obtained a poor score for Q99. WaterGAP3 performed well for all runoff signatures with the
exception of BFI, likely due to the empirical estimation of groundwater recharge and thus baseflow as a function of landscape
characteristics (Döll and Flörke, 2005). HBV-SIMREG attained high $\rho$ values for all runoff signatures.

Tables 4 and 5 also show, for the ten models and the ensembles, $\overline{OS}$ scores for the major Köppen-Geiger climate types. We
used the newly produced Köppen-Geiger climate map from Beck et al. (2016a) which is based on the high-quality WorldClim
climatic dataset (Hijmans et al., 2005) supplemented with regional climatic datasets for the USA (Daly et al., 1994), the Andes
(Manz et al., 2016), and New Zealand (Tait et al., 2006). All four LSMs (HTESSEL, JULES, ORCHIDEE, and SURFEX)
demonstrated poor performance in cold and polar climates. Conversely, PCR-GLOBWB demonstrated poor performance in
tropical, arid, and temperate climates, likely due to the overestimation of baseflow. SWBM performed well only in arid catch-
ments, at least partly due to the lack of baseflow under these conditions (Pilgrim et al., 1988; Beck et al., 2013). Similarly,

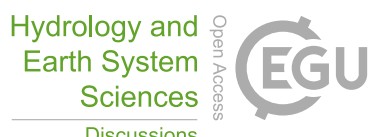

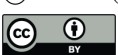

**Table 4.** For the six uncalibrated models, (i) the mean difference between simulated and observed values of the (normalized) runoff signatures ($\overline{D}$), (ii) the mean temporal correlation between simulated and observed runoff time series ($\overline{r}$), (iii) the mean overall performance in terms of runoff signatures and temporal correlation ($\overline{OS}$), and (iv) the spatial correlation between simulated and observed values of the runoff signatures ($\rho$). For each metric, the worst value is shown in orange font and the best in blue font, fading in between for intermediate values. See Figure 1 for the locations of the catchments.

| Metric | HTESSEL | JULES | ORCHIDEE | PCR-GLOBWB | SURFEX | W3RA |
|---|---|---|---|---|---|---|
| *(i) Mean difference between simulated and observed values of the (normalized) runoff signatures* | | | | | | |
| $\overline{D_{\mathrm{RC}}}$ ($n = 966$) | −0.41 | −0.16 | −0.52 | −0.02 | −0.26 | −0.21 |
| $\overline{D_{\mathrm{MAR}}}$ ($n = 966$) | −0.30 | −0.10 | −0.38 | 0.00 | −0.19 | −0.13 |
| $\overline{D_{\mathrm{T50}}}$ ($n = 966$) | −0.36 | −0.52 | 0.07 | −0.27 | −0.86 | −0.68 |
| $\overline{D_{\mathrm{BFI}}}$ ($n = 632$) | −0.06 | −1.11 | −1.22 | 1.35 | −1.37 | 0.31 |
| $\overline{D_{\mathrm{Q1}}}$ ($n = 641$) | −0.07 | 0.26 | 0.20 | −0.29 | 0.37 | −0.08 |
| $\overline{D_{\mathrm{Q99}}}$ ($n = 641$) | −0.22 | −0.73 | −0.92 | 0.27 | −0.89 | 0.08 |
| *(ii) Mean temporal correlation between simulated and observed runoff time series* | | | | | | |
| $\overline{r_{\mathrm{dly}}}$ ($n = 641$) | 0.33 | 0.23 | 0.21 | 0.34 | 0.31 | 0.44 |
| $\overline{r_{\mathrm{dly\,log}}}$ ($n = 641$) | 0.50 | 0.41 | 0.33 | 0.50 | 0.51 | 0.56 |
| $\overline{r_{\mathrm{5\,day}}}$ ($n = 641$) | 0.45 | 0.36 | 0.33 | 0.44 | 0.41 | 0.52 |
| $\overline{r_{\mathrm{mon}}}$ ($n = 966$) | 0.53 | 0.44 | 0.40 | 0.58 | 0.43 | 0.57 |
| $\overline{r_{\mathrm{mon\,clim}}}$ ($n = 966$) | 0.66 | 0.50 | 0.49 | 0.73 | 0.47 | 0.64 |
| $\overline{r_{\mathrm{yr}}}$ ($n = 966$) | 0.58 | 0.61 | 0.51 | 0.58 | 0.57 | 0.63 |
| *(iii) Mean overall performance in terms of runoff signatures and temporal correlation* | | | | | | |
| All ($n = 966$) | 0.43 | 0.39 | 0.26 | 0.41 | 0.32 | 0.46 |
| A: tropical ($n = 57$) | 0.41 | 0.46 | 0.28 | 0.03 | 0.41 | 0.39 |
| B: arid ($n = 38$) | 0.52 | 0.50 | 0.38 | 0.07 | 0.46 | 0.50 |
| C: temperate ($n = 203$) | 0.46 | 0.54 | 0.35 | 0.37 | 0.51 | 0.51 |
| D: cold ($n = 633$) | 0.43 | 0.34 | 0.23 | 0.47 | 0.25 | 0.45 |
| E: polar ($n = 35$) | 0.32 | 0.25 | 0.20 | 0.53 | 0.23 | 0.33 |
| *(iv) Spatial correlation between simulated and observed values of the runoff signatures* | | | | | | |
| $\rho_{\mathrm{RC}}$ ($n = 966$) | 0.67 | 0.64 | 0.30 | 0.56 | 0.65 | 0.60 |
| $\rho_{\mathrm{MAR}}$ ($n = 966$) | 0.80 | 0.78 | 0.61 | 0.73 | 0.79 | 0.77 |
| $\rho_{\mathrm{T50}}$ ($n = 966$) | 0.76 | 0.82 | 0.66 | 0.63 | 0.78 | 0.85 |
| $\rho_{\mathrm{BFI}}$ ($n = 632$) | 0.38 | 0.28 | 0.46 | 0.10 | 0.01 | 0.35 |
| $\rho_{\mathrm{Q1}}$ ($n = 641$) | 0.77 | 0.74 | 0.54 | 0.53 | 0.64 | 0.67 |
| $\rho_{\mathrm{Q99}}$ ($n = 641$) | 0.70 | 0.69 | 0.51 | 0.43 | 0.59 | 0.68 |
| $\rho_{\mathrm{MAR\,trend}}$ ($n = 966$) | 0.37 | 0.38 | 0.37 | 0.36 | 0.32 | 0.38 |

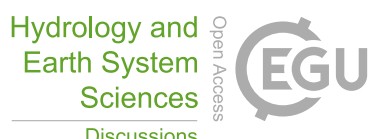

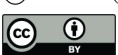

**Table 5.** For the four calibrated models and the ensembles, (i) the mean difference between simulated and observed values of the (normalized) runoff signatures ($\overline{D}$), (ii) the mean temporal correlation between simulated and observed runoff time series ($\overline{r}$), (iii) the mean overall performance in terms of runoff signatures and temporal correlation ($\overline{OS}$), and (iv) the spatial correlation between simulated and observed values of the runoff signatures ($\rho$). For each metric, the worst value is shown in orange font and the best in blue font, fading in between for intermediate values. See Figure 1 for the locations of the catchments.

| Metric | LISFLOOD | SWBM | WaterGAP3 | HBV-SIMREG | MEAN-All | MEAN-Best4 |
|---|---|---|---|---|---|---|
| *(i) Mean difference between simulated and observed values of the (normalized) runoff signatures* | | | | | | |
| $\overline{D}_{\mathrm{RC}}$ (n = 966) | 0.08 | −0.14 | −0.09 | −0.06 | −0.14 | −0.04 |
| $\overline{D}_{\mathrm{MAR}}$ (n = 966) | 0.08 | −0.07 | −0.08 | −0.03 | −0.09 | −0.02 |
| $\overline{D}_{\mathrm{T50}}$ (n = 966) | −0.13 | −0.32 | −0.34 | −0.03 | −0.31 | −0.24 |
| $\overline{D}_{\mathrm{BFI}}$ (n = 619) | 0.56 | −2.80 | −0.91 | −0.10 | −0.16 | 0.20 |
| $\overline{D}_{\mathrm{Q1}}$ (n = 641) | −0.03 | 0.76 | 0.37 | 0.12 | 0.01 | 0.00 |
| $\overline{D}_{\mathrm{Q99}}$ (n = 641) | 0.35 | −1.40 | −0.17 | −0.03 | 0.15 | 0.33 |
| *(ii) Mean temporal correlation between simulated and observed runoff time series* | | | | | | |
| $\overline{r}_{\mathrm{dly}}$ (n = 641) | 0.59 | 0.32 | 0.33 | 0.56 | 0.44 | 0.54 |
| $\overline{r}_{\mathrm{dly\,log}}$ (n = 641) | 0.70 | 0.34 | 0.56 | 0.71 | 0.64 | 0.71 |
| $\overline{r}_{\mathrm{5\,day}}$ (n = 641) | 0.64 | 0.48 | 0.52 | 0.65 | 0.59 | 0.65 |
| $\overline{r}_{\mathrm{mon}}$ (n = 966) | 0.71 | 0.63 | 0.65 | 0.74 | 0.69 | 0.72 |
| $\overline{r}_{\mathrm{mon\,clim}}$ (n = 966) | 0.84 | 0.75 | 0.76 | 0.86 | 0.80 | 0.84 |
| $\overline{r}_{\mathrm{yr}}$ (n = 966) | 0.62 | 0.60 | 0.59 | 0.62 | 0.64 | 0.63 |
| *(iii) Mean overall performance in terms of runoff signatures and temporal correlation* | | | | | | |
| All (n = 966) | 0.55 | 0.34 | 0.52 | 0.62 | 0.57 | 0.60 |
| A: tropical (n = 57) | 0.43 | 0.29 | 0.40 | 0.47 | 0.48 | 0.47 |
| B: arid (n = 38) | 0.32 | 0.44 | 0.42 | 0.55 | 0.50 | 0.44 |
| C: temperate (n = 203) | 0.52 | 0.31 | 0.48 | 0.61 | 0.59 | 0.58 |
| D: cold (n = 633) | 0.58 | 0.35 | 0.55 | 0.65 | 0.59 | 0.63 |
| E: polar (n = 35) | 0.60 | 0.25 | 0.44 | 0.60 | 0.51 | 0.57 |
| *(iv) Spatial correlation between simulated and observed values of the runoff signatures* | | | | | | |
| $\rho_{\mathrm{RC}}$ (n = 966) | 0.57 | 0.54 | 0.82 | 0.70 | 0.72 | 0.79 |
| $\rho_{\mathrm{MAR}}$ (n = 966) | 0.71 | 0.74 | 0.87 | 0.81 | 0.81 | 0.83 |
| $\rho_{\mathrm{T50}}$ (n = 966) | 0.87 | 0.88 | 0.88 | 0.91 | 0.91 | 0.90 |
| $\rho_{\mathrm{BFI}}$ (n = 619) | 0.28 | 0.37 | −0.03 | 0.71 | 0.55 | 0.54 |
| $\rho_{\mathrm{Q1}}$ (n = 641) | 0.65 | 0.73 | 0.80 | 0.76 | 0.76 | 0.78 |
| $\rho_{\mathrm{Q99}}$ (n = 641) | 0.58 | 0.09 | 0.71 | 0.76 | 0.75 | 0.74 |
| $\rho_{\mathrm{MAR\,trend}}$ (n = 966) | 0.42 | 0.39 | 0.35 | 0.37 | 0.40 | 0.39 |





Orth et al. (2015) found that SWBM performs well during dry periods for eight small Swiss catchments (60 to 392 km$^2$). LISFLOOD, W3RA, WaterGAP3, and HBV-SIMREG showed moderate to good performance for all climates.

Figure 1 presents, for the ten models and the ensembles, maps of simulated minus observed MAR for the catchments, revealing the data underlying the MAR $\overline{D}$ and $\rho$ values listed in Tables 4 and 5. Maps of all other runoff signatures are

5 presented in Supplementary material Figures S1.2–8. HTESSEL and ORCHIDEE strongly underestimate runoff for most of the catchments, while LISFLOOD appears to strongly overestimate runoff for most of the globe with the exception of snow-dominated regions. All models showed negative MAR biases in snow-dominated regions such as Alaska, the Rocky Mountains, and southern Russia, while they consistently showed positive MAR biases for the Great Plains (USA) and southern Australia. Figure 2 shows, for the ten models and the ensembles, maps of the correlation between simulated and observed monthly flows

($r_{\mathrm{mon}}$) for the catchments, showing the data underlying the $\overline{r_{\mathrm{mon}}}$ values presented in Tables 4 and 5. Maps of all other temporal variability metrics are presented in Supplementary material Figures S1.9–14. In general, LISFLOOD and HBV-SIMREG obtained high $r_{\mathrm{mon}}$ values for most catchments, while JULES, ORCHIDEE, and SURFEX obtained relatively low $r_{\mathrm{mon}}$ values for most catchments. All four LSMs showed low $r_{\mathrm{mon}}$ values for snow-dominated catchments.

Figure 3 shows, for the four models for which PET data were available, density plots of grid cell values of aridity index (AI;

ratio of long-term PET to $P$) versus runoff coefficient (RC; ratio of long-term MAR to $P$), revealing how the models behave in terms of RC under different climatic conditions. Also shown are the energy-limit line for which actual evaporation equals PET and the Budyko (1974) curve, the most well-known among several similar empirical relationships describing the competition between runoff and actual evaporation (Ol'dekop, 1911; Pike, 1964; Zhang et al., 2001; Porporato et al., 2004). Departures from the Budyko curve have been attributed to seasonality in climate, snowfall fraction of total $P$, vegetation cover, and soil

water storage (Milly, 1994; Zhang et al., 2001; Potter et al., 2005; Donohue et al., 2007; Berghuijs et al., 2014). A hydrological model would ideally produce RC values that stay above the energy-limit line and scatter around the Budyko curve. However, only W3RA appears to exhibit this behavior (Figure 3c); the other models produce RC values that deviate systematically from the Budyko curve (Figures 3a, 3b, and 3d). In addition, ORCHIDEE and WaterGAP3 produce RC values for many grid cells that fall well below the energy-limit line (Figures 3a and 3d, respectively), meaning that the actual evaporation exceeds PET

which is physically impossible. This suggests that the evaporation routines of ORCHIDEE and WaterGAP3 need to be re-evaluated. WaterGAP3 exhibits a particularly strong scatter, perhaps due to the calibration compensating for errors in the $P$, PET, or runoff data.

It is generally difficult to gain insight into why a particular model performs as it does due to the large number of interacting model components, equations, and parameters. Nevertheless, the underestimation of runoff by HTESSEL probably reflects

the excessive evaporation by HTESSEL previously reported by Haddeland et al. (2011). PCR-GLOBWB most likely suffers from suboptimal baseflow-related parameter values, since its structure is similar to that of LISFLOOD which performs markedly better. SWBM clearly suffers from the absence of a baseflow routine outside (semi-)arid regions. Although W3RA and HBV-SIMREG use an identical snow routine, W3RA performs considerably worse in snow-dominated regions, probably because HBV-SIMREG uses a snowfall gauge undercatch correction factor. The unsatisfactory performance demonstrated by

the LSMs in snow-dominated regions could be related to deficiencies in the snow routines or the energy balance estimates (see




**Figure 1.** Simulated minus observed square-root transformed mean annual runoff (MAR; units $\sqrt{\mathrm{mm\ yr^{-1}}}$) for the catchments. Each data point represents a catchment centroid ($n = 966$). Red (blue) indicates an overestimated (underestimated) MAR relative to the observations.





**Figure 2.** Correlation coefficients calculated between simulated and observed monthly runoff ($r_{mon}$; unitless) for the catchments. Each data point represents a catchment centroid ($n = 966$).




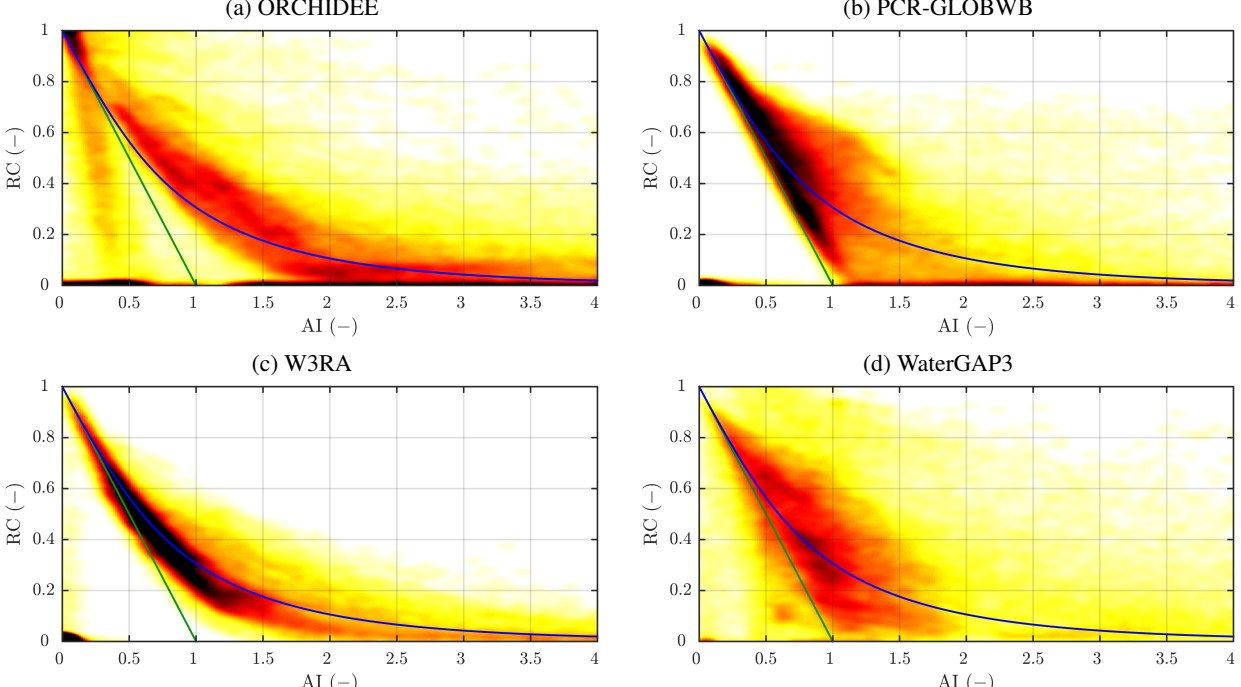

**Figure 3.** For the four models for which PET data were available, density plots of grid cell values of aridity index (AI) versus runoff coefficient (RC). The blue line represents the Budyko (1974) curve, whereas the green line represents the energy limit for which actual evaporation equals PET.

Section 4.3). WaterGAP3 and particularly HBV-SIMREG performed quite well overall, likely because of their comprehensive calibration (see Section 4.4). In any case, the pronounced inter-model performance spread found here suggests that model choice should be regarded as a critical step in any hydrological modeling study. Moreover, it underscores the importance of hydrological model uncertainty in addition to climate input uncertainty, as also emphasized in several other recent macro-scale

5    studies (Haddeland et al., 2011; Schewe et al., 2013; Prudhomme et al., 2014; Mendoza et al., 2015; Giuntoli et al., 2015a). Currently, the large majority of studies assessing the hydrological impacts of climate change completely neglect hydrological model uncertainty (Teutschbein and Seibert, 2010).

## 4.2   How well do the models perform in terms of long-term runoff trends?

The models displayed consistent MAR trends (Supplementary material Figure S1.8), meaning they respond similarly to climate

10   variability, given that none of the models account for land-use or land cover changes, urbanization, reservoir construction, or increasing atmospheric $CO_2$. However, the models obtained rather low spatial (Spearman) correlation coefficients ($\rho_{\mathrm{MAR\,trend}}$) ranging from 0.32 (SURFEX) to 0.42 (LISFLOOD; Tables 4 and 5), indicating that the simulated MAR trends correspond poorly to the observed ones. These values are lower than the (Pearson) correlation coefficients ranging from 0.52 to 0.63 obtained by Stahl et al. (2012), who evaluated MAR trends from seven models using observed runoff from 293 small Eu-



ropean catchments (100 to 1000 km$^2$), presumably due to the better quality of the European forcing and observed runoff data. Milly et al. (2005) evaluated MAR trends from a 12-model ensemble using observed runoff from 165 large catchments ($> 50\,000$ km$^2$) around the globe, obtaining a (Pearson) correlation coefficient of 0.34 which is similar to ours. These low correlations, which were somewhat unexpected given the relative ease with which MAR can be estimated (e.g., Westerberg

and McMillan, 2015; Beck et al., 2015), may be indicative of changes in non-climatic drivers of hydrological change or drift errors in the forcing or observed runoff data. We expect the inter-model variability in trends to be higher and the agreement with observations to be even lower for seasonal and monthly averages as well as runoff signatures sensitive to the shape of individual flow events (cf. Bastola et al., 2011; Gosling et al., 2011). Overall, these results suggest that studies assessing the impacts of climate change on runoff in small-to-medium sized catchments should be interpreted with considerable caution.

### 4.3 How do the results of the GHMs differ, if at all, from those of the LSMs?

Similar to Haddeland et al. (2011), the LSMs were found to produce less runoff overall (Tables 4 and 5, and Figure 1), perhaps due to their use of physically-based Richards-Darcy type equations which neglect preferential flows. We further found that the GHMs perform, on average, worse than the LSMs in rain-dominated regions: the GHMs (excluding the comprehensively calibrated models—WaterGAP3 and HBV-SIMREG; see Section 4.4) obtained mean $\overline{OS}$ scores of 0.28, 0.33, and 0.43 for

tropical, arid, and temperate climates, respectively, while the same values for the LSMs are 0.39, 0.47, and 0.47, respectively (Tables 4 and 5). However, the poor performance of the GHMs is primarily attributable to PCR-GLOBWB and SWBM. As mentioned before, PCR-GLOBWB probably suffers from a suboptimal baseflow-related parameterization, while SWBM suffers from the absence of a baseflow routine.

The GHMs do appear to perform consistently better than the LSMs in snow-dominated regions: the GHMs (again excluding

WaterGAP3 and HBV-SIMREG) obtained mean $\overline{OS}$ scores of 0.46 and 0.32 for cold and polar climates, respectively, while the same values for the LSMs are 0.31 and 0.25, respectively (Tables 4 and 5). The performance of the LSMs appears to be mainly due to a very early bias in flow timing, a very low baseflow contribution, and a misrepresentation of the seasonal cycle (Supplementary material Figures S1.4, S1.5, and S1.13, respectively). Our results are in agreement with Giuntoli et al. (2015b), who found five GHMs to outperform, on average, four LSMs using runoff observations from 252 temperate and cold catchments (64

to 1 350 000 km$^2$) located in the central USA, and with Zhang et al. (2016), who found that two LSMs performed considerably worse than two GHMs in cold and polar regions using runoff observations from 644 catchments ($> 2000$ km$^2$, upper limit not reported) around the globe. The poorer performance obtained by the LSMs is probably indicative of differences between the snow routines used by GHMs and LSMs. The GHMs use relatively simple conceptual temperature-index snow routines driven by air temperature which can be estimated with relative ease, whereas the LSMs use more complex physically-based

energy balance snow routines driven by estimates of energy balance components which are subject to considerable uncertainty, particularly in regions with complex topography (Ferguson, 1999). Although several previous studies have found that the two types of snow routines yield comparable performance (e.g., WMO, 1986; Franz et al., 2008; Zeinivand and De Smedt, 2009; Debele et al., 2010), these studies used a very small number of relatively well-instrumented catchments (six, two, one, and



three, respectively) which may have led to non-generalizable conclusions. Overall, it appears that the energy balance estimates and snow routines used by the LSMs need to be re-evaluated (cf. Zhang et al., 2016).

### 4.4 Are calibration and regionalization important or even essential?

Calibration is a prerequisite for both conceptual and physically-based hydrological models to provide reliable runoff estimates, to compensate for (i) the impossibility of measuring all required model parameters at the model application scale, (ii) lack of process understanding, (iii) possibly overly simplistic process representations, (iv) the spatio-temporal discretization of highly heterogeneous rainfall-runoff processes, and (v) errors in the forcing data (Beven, 1989; Blöschl and Sivapalan, 1995; Duan et al., 2001, 2006; McDonnell et al., 2007; Nasonova et al., 2009; Rosero et al., 2011; Minville et al., 2014). Yet, despite the development of numerous calibration techniques over the last 50 years (Dawdy and O'Donnell, 1965; Duan et al., 2004) and the current widespread availability of runoff observations (Hannah et al., 2011), macro-scale models generally tend to be uncalibrated (Sooda and Smakhtin, 2015; Bierkens, 2015; Kauffeldt et al., 2016). This is perhaps mainly due to (i) the substantial amount of work involved with calibration (e.g., Bock et al., 2015), (ii) the risk of obtaining unrealistic parameters due to equifinality and data issues (Andréassian et al., 2012), and (iii) the lack of a commonly accepted regionalization technique (Beck et al., 2016a). In addition, the modeler may feel that since their model is physically based, it does not require calibration (Beven, 1989). LSMs in particular are rarely calibrated against runoff, likely because: (i) runoff estimation is generally not among the primary aims of LSMs; (ii) for water transport in the soil, LSMs typically use Richards-Darcy type equations which are computationally expensive and require a fine vertical and temporal soil discretization; and (iii) LSMs often do not account for river routing, which limits the calibration of large catchments to longer time scales. Instead, the parameters in macro-scale models are usually based on "expert opinion" and thus founded on the bold assumption that the modeler sufficiently understands the hydrological processes, feedbacks, and parameter interactions taking place within the model for any location on Earth.

Nevertheless, out of the ten models considered in this study, four use parameters derived by calibration (LISFLOOD, SWBM, WaterGAP3, and HBV-SIMREG—all GHMs). LISFLOOD was calibrated against observed runoff for 24 large catchments (84 230 to 4 680 000 km$^2$) across the globe using the WFDEI forcing and an aggregate objective function incorporating bias, NSE, and log-transformed NSE computed from daily runoff data. The calibration might have influenced the present evaluation; although we used much smaller catchments (1000 to 5000 km$^2$), 47 % of our catchments are located within the calibration catchments. SWBM uses a spatially-uniform parameter set based on calibration using the E-OBS forcing (Haylock et al., 2008) against European data on such key hydrologic variables as soil moisture, total water storage, evaporation, and runoff (Orth and Seneviratne, 2015). For the calibration against runoff, they used observations from 436 small European catchments (mostly < 1000 km$^2$), and considered daily and monthly correlations as well as bias. The calibrated parameter set was subsequently applied globally. Besides the addition of a baseflow routine, SWBM would probably benefit from regionalized parameters that vary according to landscape characteristics. WaterGAP3 has been calibrated using the WFDEI forcing in terms of bias for the interstation regions (the catchment of a station excluding the catchments of nested upstream stations) of 2071 stations (catchment size ranging from 2830 to 966 321 km$^2$) around the globe, some of which have also been used in the current



evaluation. The calibrated parameters were subsequently regionalized to ungauged regions using multiple linear regression based on six predictors. The model does indeed perform very well for MAR and thus RC, but this did not necessarily translate into good performance for BFI (Table 5, and Figures 1 and 2). HBV-SIMREG also uses regionalized parameter fields, produced by transferring calibrated parameters from 674 small-to-medium sized "donor" catchments (10 to 10 000 km$^2$) across the globe

to "receptor" grid cells with similar climatic and physiographic characteristics (Beck et al., 2016a). Although they did not use the WFDEI forcing for the calibration, they calibrated against several of the performance metrics also used here and used 179 of our catchments as parameter donors, explaining the very good performance obtained by HBV-SIMREG (Table 5, and Figures 1 and 2).

Overall, it appears that the calibration exercises for WaterGAP3, HBV-SIMREG, and possibly LISFLOOD have resulted in

markedly improved performance. However, WaterGAP3 performed poorly in terms of $\rho_{\mathrm{BFI}}$ (Table 5), meaning the calibration of MAR did not translate into better BFI performance. These results underscore the benefits of calibrated parameters over *a priori* parameters (cf. Duan et al., 2006; Hunger and Döll, 2008; Nasonova et al., 2009; Rosero et al., 2011; Greuell et al., 2015; Zhang et al., 2016) and highlight the importance of using an objective function for the calibration that incorporates a broad range of metrics related to various important aspects of the hydrograph (cf. Gupta et al., 2008; Vis et al., 2015; Shafii and

Tolson, 2015). These results also emphasize the usefulness of regionalization techniques (Parajka et al., 2013), which typically enhance performance over the entire model domain and are thus of particular value for macro-scale modeling, given that the majority of the land surface is ungauged or poorly gauged (Sivapalan, 2003; Hannah et al., 2011). However, although there are numerous studies performing regionalization at a regional scale (see reviews by He et al., 2011; Hrachowitz et al., 2013; Razavi and Coulibaly, 2013; Parajka et al., 2013), only few studies have attempted regionalization at a macro scale (see review

by Beck et al., 2016a). We argue that more effort should be devoted to regionalizing the parameters of macro-scale models (cf. Bierkens, 2015; Döll et al., 2015).

### 4.5 What is the impact of the forcing data on the results?

There are not only strong inter-model differences in the performance patterns but also clear inter-model similarities. Specifically, all models showed negative biases in MAR in snow-dominated regions such as Alaska, the Rocky Mountains, and

25 southern Russia, while they consistently showed positive biases in MAR for the Great Plains (USA) and southern Australia (Figure 1). The high spatial correlation in the performance patterns suggests that these consistent performance patterns may be due to biases in the WFDEI $P$ data, rather than biases in the observed runoff data which are unlikely to be spatially correlated.

It is conceivable that biases are present in the WFDEI $P$ data, since the adjustment using the gauge-based CRU dataset is expected to be effective only in gauge-rich regions without complex topography. For the conterminous USA we quanti-

30 fied the biases in the WFDEI $P$ data using the high-quality Parameter-elevation Relationships on Independent Slopes Model (PRISM) climatic dataset (Daly et al., 1994), which is based on gauges and includes sophisticated corrections for undercatch and orography. Figure 4a shows the bias in mean annual $P$ from WFDEI relative to that from PRISM, suggesting that the WFDEI $P$ data are indeed subject to large biases. Figure 4b shows the bias in MAR from the MEAN-All ensemble relative to MAR from the observations, revealing a comparable bias pattern, thus confirming that the biases in the WFDEI $P$ propagate




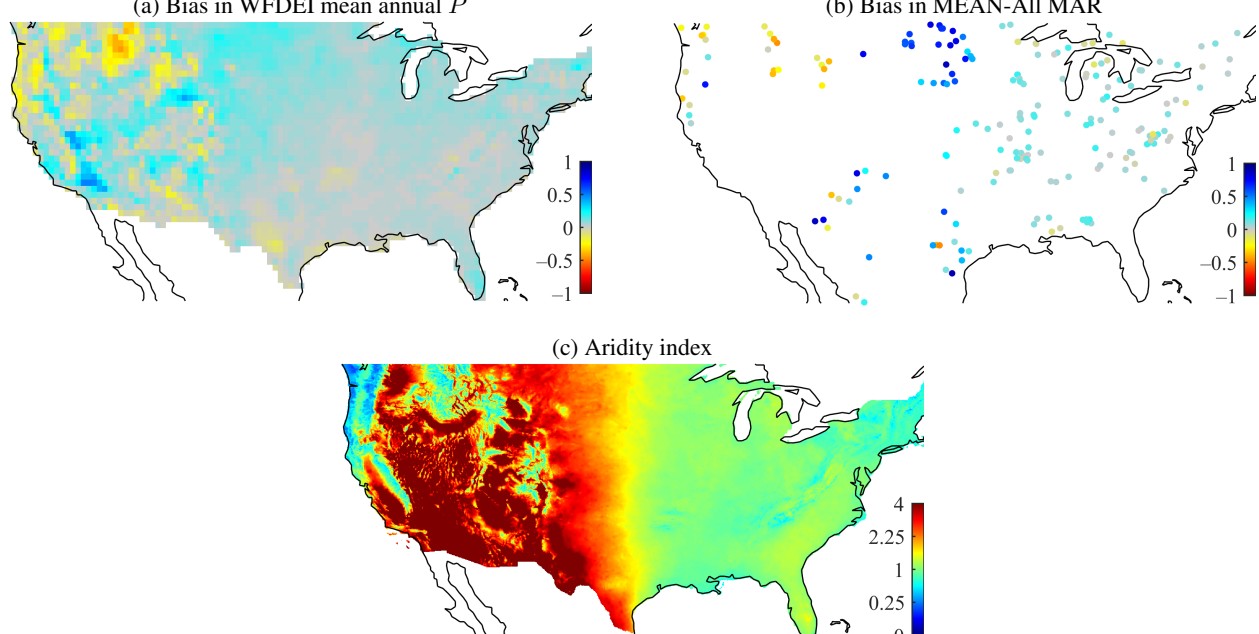

**Figure 4.** For the conterminous US, (a) the bias in mean annual $P$ from WFDEI relative to PRISM, (b) the bias in MAR from the MEAN-All ensemble relative to the observations, and (c) the aridity index, the ratio of mean annual PET (computed from PRISM air temperature using Hargreaves et al., 1985) to $P$ (PRISM; note the non-linear color scale). Each data point in panel (b) represents a catchment centroid. The bias in (a) and (b) was computed following $B = (X - R)/(X + R)$, where $B$ is the bias, $X$ the uncertain value, and $R$ the reference value. $B$ values range from $-1$ to 1. A 100 % overestimation results in $B = 1/3$, whereas a 50 % underestimation results in $B = -1/3$.

in the simulated runoff. These $P$ biases translate into even more pronounced runoff biases in (semi-)arid regions (notably the northern Great Plains; Figures 4b and 4c) due to the highly non-linear response behavior in these environments (Lidén and Harlin, 2000; Fekete et al., 2004; Van Dijk et al., 2013a). We were unable to quantify the $P$ biases globally since no other independent, global-scale $P$ dataset exists (the WorldClim and CHPclim datasets are likely to exhibit similar biases as the

5   CRU TS3.1 dataset, given that they are based on similar sets of gauges). However, we expect the $P$ biases to be at least similar, if not more severe, outside the well-instrumented conterminous USA (cf. Fekete et al., 2004; Hijmans et al., 2005; Biemans et al., 2009; Zhou et al., 2012; Kauffeldt et al., 2013; Greuell et al., 2015). It should be noted that biases in PET are probably of secondary importance as compared with biases in $P$ (Donohue et al., 2010; Sperna Weiland et al., 2011; Seiller and Anctil, 2015).

10   The global-scale quantification and reduction of these $P$ biases should be a priority for future research. Satellite-derived $P$ offers unique opportunities in this regard (e.g., Funk et al., 2015) that extend beyond the tropics with the recent launch of the Global Precipitation Measurement (GPM) mission (Smith et al., 2007). Another little-explored way of reducing $P$ uncertainty is by "doing hydrology backwards"; that is, to use information on other hydrological variables—for example, satellite-derived surface soil moisture (e.g., Brocca et al., 2014), runoff observations (e.g., Adam et al., 2006; Beck et al., 2016b), and snow-





depth observations (e.g., Cherry et al., 2005)—to reconstruct $P$ through hydrological modeling. Arguably the most important obstacles to combining multiple data sources are the inconsistent temporal coverage and scale of different data sources and the general lack of error/uncertainty estimates.

Although the models all used the same $P$ data, they used different formulations to compute PET which has likely contributed

to differences in simulated runoff among the models in energy-limited regions (Weiß and Menzel, 2008; Kingston et al., 2009; Haddeland et al., 2011; Weedon et al., 2011; Sperna Weiland et al., 2011). However, PET data were available for only four models, which is insufficient to examine whether the PET formulation has had a discernible influence on the simulated runoff, given the numerous other differences in structure and parameterization among the models.

### 4.6   How valuable are multi-model ensembles?

The multi-model ensemble MEAN-All incorporated all ten models, while MEAN-Best4 incorporated only LISFLOOD, W3RA, WaterGAP3, and HBV-SIMREG (i.e., the four models that performed best in terms of $\overline{\mathrm{OS}}$; Tables 4 and 5). MEAN-All and MEAN-Best4 were found to perform better than all individual models (with the exception of HBV-SIMREG, which has been comprehensively calibrated; Tables 4 and 5, and Figures 1 and 2). These results highlight the benefits of multi-model ensembles, in line with several previous studies (Ajami et al., 2006; Duan et al., 2007; Viney et al., 2009; Materia et al., 2010; Velázquez

et al., 2010; Gudmundsson et al., 2012; Xia et al., 2012; Yang et al., 2015). The similar $\overline{\mathrm{OS}}$ scores obtained by MEAN-All and MEAN-Best4 (0.57 and 0.60, respectively; Table 5) suggests that the inclusion of less reliable models has only limited adverse effects. It may be worthwhile for future studies to examine the benefits of more sophisticated multi-model combination techniques involving bias correction or model weighting (e.g., Ajami et al., 2006; Duan et al., 2007; Bohn et al., 2010). These weights can subsequently be transferred from gauged to ungauged areas using regionalization techniques typically used for

hydrological model parameters (Blöschl et al., 2013).

HBV-SIMREG differs from the other models because it represents a so-called "multi-parameterization ensemble", which means the model was run multiple (ten) times globally using different (regionalized) parameter sets representing different catchment response behaviors (Beck et al., 2016a). HBV-SIMREG obtained slightly better performance than both MEAN-All and MEAN-Best4 overall (Table 5), tentatively suggesting that a multi-parameterization ensemble for a single, sufficiently

flexible model provides performance comparable to a multi-model ensemble (cf. Oudin et al., 2006; Yang et al., 2011; Coxon et al., 2014). If this is confirmed, it would negate the need to set up, run, and maintain multiple models, and incentivize the development of a single community hydrological model (cf. Weiler and Beven, 2015) as well as modeling systems allowing selection of alternative model structures (cf. Bierkens, 2015), such as the Framework for Understanding Structural Errors (FUSE; Clark et al., 2008), Noah Multi-Parameterization (Noah-MP; Niu et al., 2011), and SUPERFLEX (Fenicia et al.,

2011).





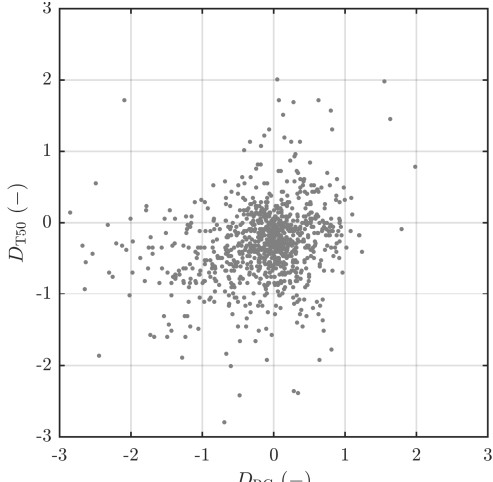

**Figure 5.** Scatterplot of the difference between simulated (MEAN-All) and observed transformed RC ($D_{RC}$) versus the difference between simulated (MEAN-All) and observed T50 ($D_{T50}$) for the catchments ($n = 966$).

### 4.7 Do all models show the early bias in runoff timing in snow-dominated catchments previously documented and what is the cause?

With the exception of ORCHIDEE and HBV-SIMREG, all models showed early T50 biases in snow-dominated regions (Supplementary material Figure S1.3), indicating that the models produce the spring snowmelt peak early, as has also been reported in several previous studies using different models and forcing data (Lohmann et al., 2004; Slater et al., 2007; Decharme and Douville, 2007; Balsamo et al., 2009; Zaitchik et al., 2010; Beck et al., 2015). The early runoff timing is probably primarily due to $P$ underestimation which leads to insufficient snow accumulation that subsequently melts too quickly (Hancock et al., 2014). Indeed, Figure 5 tentatively shows that catchments in which the models strongly underestimate runoff (i.e., negative $D_{RC}$) generally tend to exhibit an early bias in T50 (i.e., negative $D_{T50}$) and vice versa. The absence or misrepresentation of certain processes that delay snowmelt runoff in the models may have exacerbated the early runoff timing problem. Examples of such processes include the isothermal phase change of the snowpack, retainment of meltwater in the snowpack in pore spaces, infiltration of meltwater into the soil, meltwater refreezing during cold days and nights, and icejams in rivers. On the whole, more research is needed to ascertain the exact reasons of the early runoff timing.

## 5 Conclusions

The runoff estimates from ten state-of-the-art macro-scale hydrological models, all forced with the WFDEI dataset, were evaluated using runoff observations from 966 medium sized catchments around the globe. With reference to the questions posed in the introduction, the following was found:





1. The performance differed markedly among models, underscoring the importance of hydrological model uncertainty in addition to climate input uncertainty, and suggesting that model choice should be regarded as a critical step in any hydrological modeling study.

2. The models displayed similar MAR trends, although they were in poor agreement with observed trends. Model-based runoff trends in small-to-medium sized catchments should thus be interpreted with considerable caution.

3. Considering only the uncalibrated models, the GHMs performed similarly to the LSMs in rainfall-dominated regions but consistently better than the LSMs in snow-dominated regions, perhaps due to the use of more data-demanding snow routines or the misrepresentation of frozen-soil and snowmelt processes by the LSMs.

4. The models that have been calibrated obtained higher scores for the performance metrics incorporated in the respective objective functions used for calibration, suggesting that a broad range of performance metrics should be incorporated in the objective function. Overall, more effort should be devoted to calibrating and regionalizing the parameters of macro-scale models.

5. The WFDEI $P$ forcing data still appear to contain substantial biases, despite adjustments using gauge observations. These $P$ biases translate into biases in the simulated runoff which are amplified in (semi-)arid regions. In snow-dominated regions there appears to be a consistent underestimation in $P$ and thus simulated runoff.

6. The multi-model ensembles obtained only slightly worse performance than the best (calibrated) model, and the inclusion of less reliable models did not severely degrade the performance. A multi-parameterization ensemble for a single, sufficiently flexible model is easier to realize but we speculate may yield the same performance benefits as a multi-model ensemble.

7. Most models were indeed found to generate the spring snowmelt peak early, probably due to the previously mentioned $P$ underestimation and the absence or misrepresentation of certain processes that delay snowmelt runoff in the models.

*Author contributions.* H.B. designed and performed the model evaluation and wrote most of the manuscript. A.v.D., A.d.R., E.D., G.F., R.O., and J.S. helped with the interpretation of the results and the writing of the manuscript. H.B., A.v.D., A.d.R., E.D., G.F., and R.O. assisted in running the hydrological models and making available the model output.

*Acknowledgements.* The Global Runoff Data Centre (GRDC) and the U.S. Geological Survey (USGS) are thanked for providing most of the observed runoff data. We gratefully acknowledge the modeling groups participating in the eartH2Observe project for providing the simulated runoff data. This research received funding from the European Union Seventh Framework Programme (FP7/2007–2013) under grant agreement no. 603608, "Global Earth Observation for integrated water resource assessment": eartH2Observe. The views expressed herein are those of the authors and do not necessarily reflect those of the European Commission.



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
