# Peer review of "Global evaluation of runoff from ten state-of-the-art hydrological models"

_Hydrology and Earth System Sciences, 2016_

## Referee Comment (RC1) · L. Gudmundsson (Referee) · 5 Jul 2016

OVERALL RATING:

The paper presented by Beck et al is concerned with the tedious but important task of model evaluation. Overall the paper is interesting in scope, well written and the results are clearly presented. Consequently, I do definitely support the publication of the presented work.

Nevertheless, I do have several comments/suggestions which the authors may wish to consider prior to the publication of the manuscript in a final form. For the sake of clarity, I do list "specific comments" and "small comments" below.

SPECIFIC COMMENT:

[Figure]

** Specific Comment 1: ** While the paper is generally clearly written and most of the conclusions are supported by quantitative evidence there is a tendency for value statements (e.g. p. 2, l. 26: "NSE . . . is [a] . . . flawed metric"), claims (e.g. p. 1, l. 11: ". . . more effort should be devoted to calibration. . .") or speculations (e.g. p. 9, l. 32: ". . .performed well. . . due to the lack of baseflow. . ."), which are not clearly highlighted as the authors interpretations or opinions. Although I do value if researchers defend their views on specific topics, I do also belief that it is important to clearly separate "hard facts" (either theoretical or quantitative) from soft interpretation and opinions in a scientific text. Therefore, I would like to encourage the author team to carefully revise the text of the manuscript, aiming at separating opinions from facts that are supported by either theory or quantitative analysis.

** Specific Comment 2: ** I do highly value the analysis presented in Figure 3, as this is a compelling way to investigate the physical consistency of the considered models with respect to the coupled water and energy balance. Unfortunately, the authors did only conduct this analysis for four models that output potential evapotranspiration, $E_p$, which is used to compute the aridity index.

An alternative approach, which was actually used by Budyko (1974), is to compute the aridity index as the ratio $\lambda Rn/P$ where P is precipitation, Rn net-radiation and $\lambda$ the latent heat of vaporization. This has the advantage that the results are strictly interpretable in the context of the coupled energy and water balance. In addition, this would allow to evaluate the output of all considered models.

In addition, I would like to question the authors conclusion that the models are most realistic if they scatter around the Budyko-curve (ref. p. 12, l. 20). In fact, Budyko (1974) developed the curve on the basis of a limited number of catchment observations and it is a-priori not clear whether models need to be close to this empirical rule. I do, however, fully support the authors conclusions that data-points that are outside the energy and water supply limits are a strong indication for issues in the model physics.
Finally, the similarity Figure 3 with the work of Greve et al (2014, doi: 10.1038/ngeo2247), who used the budyko framework to evaluate the credibility of reconstructions of E, P and Ep and Greve et al (2015, doi: 10.1002/2015GL063449, and sorry for citing myself), who introduced a formal way to account for scatter in the Budyko-space caught my attention. Although I am not sure if this is beneficial in the context of the presented paper, I could imagine that the tools provided in both mentioned studies might be helpful do develop additional quantitative insights to model performance.

SMALL COMMENTS:

Page 1, line 8-9: Repeated use of "(uncalibrated)". I assume one should be calibrated

Table 1: The way the authors formulate the description of Table 1 reads as this would be a comprehensive review of model validation studies. I am, however, aware of at least two further studies – again, sorry for citing myself - (Gudmundsson et al 2012, doi: 10.1029/2011WR010911, Gudmundsson & Seneviratne 2015, doi: 10.5194/hess-19-2859-2015), that conduct similar assessments. Therefore, I would encourage the authors to either emphasise that the list of studies mentioned in Table 1 is not comprehensive, or to provide a more systematic summary of previous assessments.

Page 4, line 10: "... the combination of Penman-Monthie equations..." reads strange.

Page 6, line 10: To me it is not clear why the square-root transform is necessary, please explain.

Page 6, line 10: Is Q1 (Q99) a very high or a very low value. In hydrology both definitions are used. Please specify

Equation (1) and associated text: Why do you not use the observations to determine $\sigma_o$? To me this would be much more intuitive and would help to avoid the usage of another dataset which is prone to estimation uncertainty.

Table 3, line 1: Which P dataset was used? Is it the same that was used to drive the

models or another one?

Tables 4 & 5: I do like the detailed information, but it would be much more accessible if it could be presented in figures (e.g. bar-plots)

Figure 3: Colour scale for the density is missing.

———————————————

---

## Referee Comment (RC2) · Anonymous Referee #2 · 8 Jul 2016

General comment The manuscript is to some extent a sequel to the paper Beck et al. (2016) published in Water Resources Research; in both papers, the performance of global hydrological models (those that are included in a research project) is evaluated against time series of streamflow in small basins with areas of less than 5,000 km2, e.g. less than two (out of globally 67000) 0.5° grid cells. In the submitted manuscript, the number of performance indicators has been increased and the impact of forcing data on results has additionally been investigated. While there is some added value to this, the results obtained in Beck et al. (2016) have, in my opinion, not been sufficiently used in designing the study and writing the new manuscript, and conclusions are not well founded. My major concern is that given the overall poor capability global-scale models for estimating runoff in small basins (given e.g. the uncertainty in climate data), which however is not clearly shown in the manuscript but in Beck et al. (2016), the quality of

the models even with respect to runoff generation cannot be compared well with the selected evaluation approach (streamflow in small upstream basins) (See major point 3). At least this problem has to be clearly shown and discussed. In addition, there are various points that need clarification.

Specific comments

1) One major conclusion of the manuscript is that "more effort should be devoted on calibrating and regionalizing the parameters of macro-scale models" which the authors base on the fact that the four models that are calibrated (in very different ways) show better values for the selected performance indicators than the six non-calibrated models. However, the model comparison in Beck et al. (2016), which included not only the calibrated/regionalized version of the model HBV-SIMREG but also a version where all 14 model parameters were globally uniform (not even considering independent land cover and soil information, e.g. rooting depth), showed that this model version has a better performance than all or most (depending on metric) of the other more complex calibrated or non-calibrated models (comp. Tables 7 and 8 of Beck et al. 2016). I therefore suggest to include the HBV-SIMREG version with spatially uniform parameters into the analysis. HBV-SIMREG runoff that is computed as an ensemble mean of 10 model runs with different parameters sets. Then, conclusions regarding the benefits of calibration/regionalization of should be formulated more carefully.

2) The study of Beck et al. (2016) also indicates that performance of the HBV-SIMREG model results that are not derived as the ensemble mean of 10 runs with 10 different parameter sets but just 1 (derived from the most similar donor catchment) perform only slightly worse than the ensemble mean and better than the other models (Tables 6 and 7 of Beck et al.). Therefore, the conclusion that the fact that HBV-SIMREG with 10 runs performs better than the ensemble mean of all models tentatively suggests that a multi-parameterization ensemble for a single, sufficiently flexible model could replace multi-model ensemble studies (p. 20), is not backed by the analysis in the manuscript. I suggest including the HBV-SIMREG variant with 1 run/parameter set

only, and consider the result when formulating such a conclusion. In addition, it should be taken into account (and explained very clearly in the manuscript) that HBV-SIMREG only computes runoff in 0.5° grid cells and not river discharge, as grid to grid lateral routing including the impact of lakes and wetlands as well as water abstraction are not simulated by this model. I suggest adding a table in which the scope of the different models as well as well as the specific calibration/regionalization approach (including number of adjusted parameters) are listed (in section 2). And to clearly state that global hydrological models currently cannot and do not aim at representing reality at scales below 5000 km2.

3) The third information of Beck et al. (2016) that has not been made good use of for at least framing the extensive performance comparison done in the submitted manuscript is the information provided on Nash-Sutcliffe efficiency NSE for the 10 models (Table 8 in Beck et al. 2016). Different from the aggregate objective function AOF, the well-known NSE allows the reader to understand the overall very poor performance of all global models at the scale of small basins. For example, daily NSE of all 10 models (for 1113 catchments) was negative for all models and the ensemble mean (so mean discharge would have been a better estimator than the models, while monthly NSE varied between -1.16 and 0.17! Therefore, information on the NSE should be added. However, it appears to me that one cannot really say that a model achieving a NSE of e.g. 0.17 is better than a model achieving a NSE of -0.17, they are both just very poor predictors. So what can we really learn from the comparison? This has to be deduced more carefully. I would also suggest to add to the supplement the hydrographs of e.g. 4 selected calibration basins (observed and 10 model results) and a table with the pertaining performance indicators (like Table 5) so that the reader can see the meaning of these performance indicators.

4) The conclusions regarding underestimation of snow precipitation need to be better supported. You should take into account that WFDEI precipitation includes an under-catch correction. However, for the USA, WFDEI mostly overestimates PRISM precipitation ("with sophisticated corrections for undercatch) and mean runoff. Maybe the WFDEI undercatch correction is overestimated in the USA and underestimated elsewhere? Maybe you could analyse the uncorrected precipitation data that went into WFDEI and see if they are already higher than the PRISM values. You should also discuss why two models do not show the early bias. One of the adjusted HBV-SIMREG parameters is snow undercatch, which may take care of this problem, but does it? And what may be the reason in case of ORCHIDEE?

5) Try to explain more the behavior of the different models (to the extent this is possible)

Other/technical comments

P1L1: Replace "runoff" by "streamflow" P6L14: I find the mean (over 966 or 641 basins) difference between simulated and observed runoff signature D not very informative and suggest adding the standard deviation of this differences in Tables 4 and 5. Maybe do this also to the temporal correlations. P7L10: I would not say that the Spearman rank correlation coefficients evaluate the ability to simulate "the spatial variability" but just "the variability among the observation basins". P12L25: AET in WaterGAP can exceed PET due to calibrating against mean annual discharge; while this may be unphysical, it may correct for a wrong PET estimates. So it is not the evapotranspiration routines that need to be re-evaluated but the PET (or P) estimates. P17L34: How many basins coincide? Fig. 5: Use color to indicate snow-dominated catchments and/or to color by latitude.

---

## Author Comment (AC1) · 20 Aug 2016

**Discussion comment 1**

OVERALL RATING: The paper presented by Beck et al is concerned with the tedious but important task of model evaluation. Overall the paper is interesting in scope, well written and the results are clearly presented. Consequently, I do definitely support the publication of the presented work.

Nevertheless, I do have several comments/suggestions which the authors may wish to consider prior to the publication of the manuscript in a final form. For the sake of clarity, I do list "specific comments" and "small comments" below.

We would like to thank Dr. Gudmundsson for his positive remarks, thorough review, and useful remarks. Below we respond to each of his comments in green font.

SPECIFIC COMMENT:

** Specific Comment 1: ** While the paper is generally clearly written and most of the conclusions are supported by quantitative evidence there is a tendency for value statements (e.g. p. 2, l. 26: "NSE . . . is [a] . . . flawed metric"), claims (e.g. p. 1, l. 11: ". . . more effort should be devoted to calibration. . .") or speculations (e.g. p. 9, l. 32: ". . .performed well. . . due to the lack of baseflow. . ."), which are not clearly highlighted as the authors interpretations or opinions. Although I do value if researchers defend their views on specific topics, I do also belief that it is important to clearly separate "hard facts" (either theoretical or quantitative) from soft interpretation and opinions in a scientific text. Therefore, I would like to encourage the author team to carefully revise the text of the manuscript, aiming at separating opinions from facts that are supported by either theory or quantitative analysis.

We agree and have therefore re-read the text with this in mind. The sentence "the Nash and Sutcliffe (1970) efficiency (NSE), which is increasingly considered to be a flawed metric for model performance" was changed to "the Nash and Sutcliffe (1970) efficiency (NSE), which has been criticized in several previous studies". The sentence "SWBM performed well only in arid catchments, at least partly due to the lack of baseflow under these conditions" was changed to "SWBM performed well only in arid catchments, probably at least partly due to the lack of baseflow under these conditions".

** Specific Comment 2: ** I do highly value the analysis presented in Figure 3, as this is a compelling way to investigate the physical consistency of the considered models with respect to the coupled water and energy balance. Unfortunately, the authors did only conduct this analysis for four models that output potential evapotranspiration, Ep, which is used to compute the aridity index.

Thank you for the compliment. For Tier-2 of the eartH2Observe project we hope that all modeling groups release their $E$p and $R$n data.

An alternative approach, which was actually used by Budyko (1974), is to compute the aridity index as the ratio λRn/P where P is precipitation, Rn net-radiation and λ the latent heat of vaporization. This has the advantage that the results are strictly interpretable in the context of the coupled energy and water balance. In addition, this would allow to evaluate the output of all considered models.

This is a very good suggestion. We have re-done Figure 3 using the ratio $\lambda Rn/P$ for the models without $E$p data. We have also changed the text to discuss the results of the added models.

In addition, I would like to question the authors conclusion that the models are most realistic if they scatter around the Budyko-curve (ref. p. 12, l. 20). In fact, Budyko (1974) developed the curve on the basis of a limited number of catchment observations and it is a-priory not clear whether models need to be close to this empirical rule. I do, however, fully support the authors conclusions that data-points that are outside the energy and water supply limits are a strong indication for issues in the model physics.

So far nearly all (large-scale) studies have shown that observations scatter around the Budyko curve or similar curves. In light of this, we argue that, for a model to be considered an accurate representation of reality, it should exhibit a similar pattern. Note that we are not saying or implying that the models should scatter *closely* to the Budyko curve.

Finally, the similarity Figure 3 with the work of Greve et al (2014, doi: 10.1038/ngeo2247), who used the budyko framework to evaluate the credibility of reconstructions of E, P and Ep and Greve et al (2015, doi: 10.1002/2015GL063449, and sorry for citing myself), who introduced a formal way to account for scatter in the Budyko-space caught my attention. Although I am not sure if this is beneficial in the context of the presented paper, I could imagine that the tools provided in both mentioned studies might be helpful do develop additional quantitative insights to model performance.

Thank you for bringing these two papers to our attention. We could indeed use the approach of Greve et al. (2015) to quantify the deviation of the data points from the Budyko curve. However, we feel that summary statistics are not necessary since even a quick glance at the density plot already provides a wealth of (qualitative) information about the model behavior. Moreover, it is not our objective to quantify exactly how well each model "follows" the Budyko curve, since the Budyko curve is, after all, an empirically fitted equation (as recognized by the reviewer in the preceding comment).

SMALL COMMENTS:

Page 1, line 8-9: Repeated use of "(uncalibrated)". I assume one should be calibrated

To wanted to underscore that we refer in both cases to the uncalibrated models. We have removed the parentheses to improve the readability.

Table 1: The way the authors formulate the description of Table 1 reads as this would be a comprehensive review of model validation studies. I am, however, aware of at least two further studies – again, sorry for citing myself - (Gudmundsson et al 2012, doi: 10.1029/2011WR010911, Gudmundsson & Seneviratne 2015, doi: 10.5194/hess-19-2859-2015), that conduct similar assessments. Therefore, I would encourage the authors to either emphasise that the list of studies mentioned in Table 1 is not comprehensive, or to provide a more systematic summary of previous assessments.

Thanks for mentioning these two studies, we have added them to Table 1. In addition, we have added "to the best of our knowledge" to the Introduction and the caption of Table 1, to highlight that, while we made our best effort to find all studies, some may still be missing.

Page 4, line 10: ". . . the combination of Penman-Monthie equations. . ." reads strange.

Changed to "the Penman-Monteith combination equation", the more common name used in the hydrological literature.

Page 6, line 10: To me it is not clear why the square-root transform is necessary, please explain.

The square-root transformation was necessary to give more weight to small values of the signatures (see page 6 lines 12–13 of the original m/s), i.e., to make the $D$ values from catchments in different climatic zones to be more similar in magnitude and thus more intercomparable. Without the square-root transformation of, for example, the mean annual runoff (MAR) values, the corresponding average deviation ($D$) values would be dominated by tropical catchments, which tend to exhibit very high runoff amounts.

Page 6, line 10: Is Q1 (Q99) a very high or a very low value. In hydrology both definitions are used. Please specify Equation (1) and associated text: Why do you not use the observations to determine σo? To me this would be much more intuitive and would help to avoid the usage of another dataset which is prone to estimation uncertainty.

Table 3 specifies that Q1 and Q99 are *exceedance* percentiles related to peak and low flows, respectively. We have added "exceedance" also to the main text to avoid confusion.

Our reason for using the fully global signature maps from the Global Streamflow Characteristics Dataset (GSCD) is because it provides a much more representative global picture than a sparse, unevenly distributed set of observations would. Note that the GSCD dataset is completely observation-driven.

Table 3, line 1: Which P dataset was used? Is it the same that was used to drive the models or another one?

For calculating the RC we used $P$ data from the WFDEI dataset, which has also been used to drive each of the models.

Tables 4 & 5: I do like the detailed information, but it would be much more accessible if it could be presented in figures (e.g. bar-plots)

We appreciate the suggestion, but the drawback of bar plots is that they make it difficult to deduce the actual values. On the other hand, the current table provides the actual values and allows readers to quickly interpret the results using the colors. However, if the editor feels that bar plots are more appropriate we can certainly make this change.

Figure 3: Colour scale for the density is missing.

Thank you for the comment, we have added a color scale.

---

## Author Comment (AC2) · 24 Aug 2016

**Discussion comment 2**

General comment The manuscript is to some extent a sequel to the paper Beck et al. (2016) published in Water Resources Research; in both papers, the performance of global hydrological models (those that are included in a research project) is evaluated against time series of streamflow in small basins with areas of less than 5,000 km2, e.g. less than two (out of globally 67000) 0.5◦ grid cells. In the submitted manuscript, the number of performance indicators has been increased and the impact of forcing data on results has additionally been investigated. While there is some added value to this, the results obtained in Beck et al. (2016) have, in my opinion, not been sufficiently used in designing the study and writing the new manuscript, and conclusions are not well founded. My major concern is that given the overall poor capability global-scale models for estimating runoff in small basins (given e.g. the uncertainty in climate data), which however is not clearly shown in the manuscript but in Beck et al. (2016), the quality of the models even with respect to runoff generation cannot be compared well with the selected evaluation approach (streamflow in small upstream basins) (See major point 3). At least this problem has to be clearly shown and discussed. In addition, there are various points that need clarification.

We sincerely thank the reviewer for his comments and are glad that he/she believe the study provides added value.

We do not fully agree that the "poor" (a very subjective term) capability of the models is "not clearly shown". Quite the contrary, in fact: we tried to be as transparent as possible in presenting the performance scores by explicitly showing the results for all five signatures and all six correlation coefficients for all models and all catchments (see Tables 4 and 5 and Figures 1 and 2 in the m/s, and Figures 1–14 in the Supplementary material). We prefer to leave it to the reader to determine whether this constitutes poor performance.

In describing the models' performance as 'poor', the reviewer may be drawing a comparison between global models and locally-calibrated hydrological models in such small catchments. While potentially correct on average (in individual cases it would surely depend on the quality of model, forcing and calibration), we feel that this is not the main insight to be gained from our analysis since (i) these local models are only available in certain regions and thus not a viable choice for researchers interested in large-scale hydrological simulations in predominantly ungauged regions, and (ii) this is already widely documented (we provided references in the text, see page 17 lines 7–8).

Specific comments

1) One major conclusion of the manuscript is that "more effort should be devoted on calibrating and regionalizing the parameters of macro-scale models" which the authors base on the fact that the four models that are calibrated (in very different ways) show better values for the selected performance indicators than the six non-calibrated models. However, the model comparison in

Beck et al. (2016), which included not only the calibrated/regionalized version of the model HBV-SIMREG but also a version where all 14 model parameters were globally uniform (not even considering independent land cover and soil information, e.g. rooting depth), showed that this model version has a better performance than all or most (depending on metric) of the other more complex calibrated or non-calibrated models (comp. Tables 7 and 8 of Beck et al. 2016). I therefore suggest to include the HBV-SIMREG version with spatially uniform parameters into the analysis. HBV-SIMREG runoff that is computed as an ensemble mean of 10 model runs with different parameters sets. Then, conclusions regarding the benefits of calibration/regionalization of should be formulated more carefully.

While we can see the value of such an analysis, we could not include HBV with spatially-uniform parameters in the current m/s since it is not part of the eartH2Observe collection of models (noting that the objective of the m/s is to evaluate the eartH2Observe collection of models).

Table 7 of Beck et al. (2016) shows that HBV with spatially-uniform parameters performs overall worse than two models but better than seven models. Thus, while HBV with spatially-uniform parameters performs indeed quite well among the models, it certainly did not perform beyond the range of the other models. Accordingly, the fact that HBV-SIMREG outperforms the other models is really mainly attributable to the calibration and regionalization, and our conclusion would not change if we were to include HBV with spatially-uniform parameters in the current analysis.

2) The study of Beck et al. (2016) also indicates that performance of the HBV-SIMREG model results that are not derived as the ensemble mean of 10 runs with 10 different parameter sets but just 1 (derived from the most similar donor catchment) perform only slightly worse than the ensemble mean and better than the other models (Tables 6 and 7 of Beck et al.). Therefore, the conclusion that the fact that HBV-SIMREG with 10 runs performs better than the ensemble mean of all models tentatively suggests that a multi-parameterization ensemble for a single, sufficiently flexible model could replace multi-model ensemble studies (p. 20), is not backed by the analysis in the manuscript. I suggest including the HBV-SIMREG variant with 1 run/parameter set only, and consider the result when formulating such a conclusion.

The reviewer suggests that our *speculation* that multi-parameterization models may substitute multi-model ensembles is not supported by the results. We certainly agree that the results here do not unequivocally support our speculation and have therefore used cautious terms like "may", "tentatively", "if this is confirmed", and "speculate". However, our conclusion that HBV-SIMREG with 10 runs performs better than the ensemble mean is certainly not unfounded. The good performance of HBV-SIMREG is certainly the combined effect of the calibration/regionalization and the use of ensembles, as Beck et al. (2016) demonstrate unambiguously. We prefer to retain the statement that multi-parameterization models *may* substitute multi-model ensembles, as we hope to encourage research and development in this very promising direction.

In addition, it should be taken into account (and explained very clearly in the manuscript) that HBV-SIMREG only computes runoff in 0.5◦ grid cells and not river discharge, as grid to grid lateral

routing including the impact of lakes and wetlands as well as water abstraction are not simulated by this model. I suggest adding a table in which the scope of the different models as well as well as the specific calibration/regionalization approach (including number of adjusted parameters) are listed (in section 2).

We thank the reviewer for the suggestion. In the m/s we state that "three of the models did not not simulate river routing (JULES, SWBM, and HBV-SIMREG)" (page 5 line 5). HBV-SIMREG is thus not the only model without a routing routine. However, whether the models have a routing routine is less relevant given that we analyze daily non-routed specific runoff (mentioned on page 4 lines 14–15). Besides HBV-SIMREG, four other models also do not account for lakes, while five other models also do not account for water use. However, this is also less relevant, as streamflow observations for catchments with intensive irrigation and/or large reservoirs were excluded (page 5 lines 9–12).

A comprehensive table with model specifics can be found in Dutra et al. (2015). Since the current m/s is already quite long we prefer not to repeat all this information, but refer to Dutra et al. (2015) instead.

And to clearly state that global hydrological models currently cannot and do not aim at representing reality at scales below 5000 km2.

While we cannot speak for all model developers, some of the models have definitely been designed to represent reality at scales <5000 km$^2$ (e.g., LISFLOOD, WaterGAP3, and W3RA, which were run at 0.1°, 0.08°, and 0.05° spatial resolutions, respectively). Furthermore, the results demonstrate that some of the models indeed do represent reality to a certain degree at these scales. LISFLOOD and HBV-SIMREG, for example, exhibit high monthly correlations (>0.8) for the large majority of the catchments (Figure 2), indicating that they simulate monthly flows with satisfactory skill at scales <5000 km$^2$.

Given the overbearing importance of precipitation in predicting runoff, we would expect the ability to represent reality at smaller scales depends more on the precipitation forcing than on the models per se, and indeed there is considerable published evidence for this.

3) The third information of Beck et al. (2016) that has not been made good use of for at least framing the extensive performance comparison done in the submitted manuscript is the information provided on Nash-Sutcliffe efficiency NSE for the 10 models (Table 8 in Beck et al. 2016). Different from the aggregate objective function AOF, the wellknown NSE allows the reader to understand the overall very poor performance of all global models at the scale of small basins. For example, daily NSE of all 10 models (for 1113 catchments) was negative for all models and the ensemble mean (so mean discharge would have been a better estimator than the models, while monthly NSE varied between -1.16 and 0.17! Therefore, information on the NSE should be added. However, it appears to me that one cannot really say that a model achieving a NSE of e.g. 0.17 is better than a model achieving a NSE of -0.17, they are both just very poor predictors. So

what can we really learn from the comparison? This has to be deduced more carefully. I would also suggest to add to the supplement the hydrographs of e.g. 4 selected calibration basins (observed and 10 model results) and a table with the pertaining performance indicators (like Table 5) so that the reader can see the meaning of these performance indicators.

As requested, NSE scores for the models have been added to the Supplementary material. In addition, we have added hydrographs including statistics for four catchments. With respect to NSE scores, we note that this measure does not provide an adequate summary of overall model performance. The NSE has been criticized in several previous studies, as has been explicitly mentioned in the current m/s (page lines 25–26) as well as in Beck et al. (2016), for being overly sensitive to peak flows. For this reason Beck et al. (2016) did not use the NSE for the main analysis, but rather an aggregate performance metric very similar to the one used here.

If the reviewer argues that a NSE of 0.17 is not better than an NSE of –0.17 than we take this to mean that the reviewer also does not see NSE as a good measure of model performance.

4) The conclusions regarding underestimation of snow precipitation need to be better supported. You should take into account that WFDEI precipitation includes an undercatch correction. However, for the USA, WFDEI mostly overestimates PRISM precipitation ("with sophisticated corrections for undercatch) and mean runoff. Maybe the WFDEI undercatch correction is overestimated in the USA and underestimated elsewhere? Maybe you could analyse the uncorrected precipitation data that went into WFDEI and see if they are already higher than the PRISM values. You should also discuss why two models do not show the early bias. One of the adjusted HBV-SIMREG parameters is snow undercatch, which may take care of this problem, but does it? And what may be the reason in case of ORCHIDEE?

Thank you for the suggestions. We have added the following text to Section 4.3 in an effort to better highlight the sources of uncertainty in the WFDEI $P$ data: "It is conceivable that biases are present in the WFDEI $P$ data, because: (i) the CRU dataset, which has been used to correct the WFDEI dataset, is based on only a subset of the available gauges and does not explicitly account for orographic effects; (ii) in sparsely gauged regions the correction using CRU is more likely to deteriorate rather than improve the $P$ estimates; and (iii) the Adam and Lettenmaier (2003) gauge undercatch correction factors are based on interpolation of a very sparse sample of gauges and thus subject to considerable uncertainty." We now also mention that PRISM is based on considerably more stations than CRU.

The snowfall correction parameter of HBV-SIMREG could indeed be responsible for the better runoff timing, we appreciate the suggestion. We have added the following text to Section 4.7: "The fact that HBV-SIMREG performs well in this regard is probably attributable to the snowfall gauge undercatch correction factor of the model." For ORCHIDEE we are unsure about the cause for the good runoff timing.

5) Try to explain more the behavior of the different models (to the extent this is possible)

With the current results we cannot draw further robust conclusions regarding each model's behavior. A detailed evaluation of each model independently would allow such a discussion, but this is beyond the scope of the current m/s.

Other/technical comments

P1L1: Replace "runoff" by "streamflow"

Done. Thanks for the suggestion.

P6L14: I find the mean (over 966 or 641 basins) difference between simulated and observed runoff signature D not very informative and suggest adding the standard deviation of this differences in Tables 4 and 5. Maybe do this also to the temporal correlations.

The mean $D$ provides information about the average deviation between simulated and observed signature values. We feel this information is very useful to identify, for example, if a particular model tends to produce insufficient or excessive baseflow (using the BFI signature), or if it consistently under- or overestimates peak flows (using the Q1 signature).

We prefer not to include standard deviations. First, they are difficult to interpret, being influenced by the simulated as well the observed signature distributions. Second, they require the distribution of $D$ values to be (more or less) normally shaped, which they are not. Finally, they would clutter the tables and deteriorate the readability.

P7L10: I would not say that the Spearman rank correlation coefficients evaluate the ability to simulate "the spatial variability" but just "the variability among the observation basins".

Thanks for the suggestion, we have made the change.

P12L25: AET in WaterGAP can exceed PET due to calibrating against mean annual discharge; while this may be unphysical, it may correct for a wrong PET estimates. So it is not the evapotranspiration routines that need to be re-evaluated but the PET (or P) estimates.

Thank you for pointing this out. We agree and have changed the text accordingly.

P17L34: How many basins coincide?

This information is unfortunately not available.

Fig. 5: Use color to indicate snow-dominated catchments and/or to color by latitude.

Thank you for the comment, we have added a color scale to the figure.

---

## Author Response (AR1)

Dear Editor,

We are hereby submitting a revision of our m/s. We are grateful for your handling of the manuscript, and would like to extend our appreciation to the three reviewers as their comments helped to improve the m/s.

Key changes to our revised m/s include:
1. Added a table with qualitative interpretations of intervals of the performance metrics (Table 4) and changed the text accordingly.
2. Introduced a discrete coloring of the results table that reflects the qualitative interpretations (Table 5).
3. Merged the two results tables (Tables 5 and 6 in the original m/s) to save space (Table 5 in the revised m/s).
4. Added three more studies to Table 1.
5. Explained why the WFDEI data are likely to contain biases (P19L4-8).
6. Produced Budyko density plots for three additional models using net radiation to compute the aridity index (see Figure 3).
7. Added a color scale to the Budyko density plots (see Figure 3).
8. Added NSE scores to the Supplementary information and discussed them in the revised m/s (P13L33-P14L7).
9. Added some text to put the results of Beck et al. (2016) in the context of the present results (P18L13-14).

Below in green a point-by-point response to each of the comments.

Yours truly,
Hylke Beck (on behalf of all co-authors)

**Editor Decision: Reconsider after major revisions** (06 Sep 2016) by Stan Schymanski
Comments to the Author:
Dear authors,

The two reviewers made a number of valid points, for which I am very thankful. Unfortunately, the third reviewer did not find time to review the paper. It appears that the paper is of interest, but has a general deficiency that motivated a few of the points of critique raised by the reviewers. This is the lack of clarity about what is considered satisfactory model performance and what the data actually says, independently of the authors' opinion. Reviewer #1 criticised that hard facts are not sufficiently separated from opinions, whereas Reviewer #2 disagreed on several occasions with the authors on what may be considered satisfactory model performance. I concur with both reviewers and I believe that the paper is only publishable in HESS if these points are clarified, if facts are clearly separated from opinions and if all conclusions are supported by verifiable facts. I suggest to either separate the discussion from the results section, or to more clearly separate opinions and recommendations from the presentation of the results and hard facts in the "Results and discussion" section. In addition, I suggest for the conclusions section to limit the list of conclusions to those that are directly supported by the evidence presented in this paper, which may be followed by a paragraph of a more speculative character, clearly indicating that this reflects the authors' opinion.

We have made additional efforts to separate opinion from fact throughout the m/s. For example, we deleted the statements *"suggesting that a broad range of performance metrics should be incorporated in the objective function"* and *"more effort should be devoted to calibrating and regionalizing the parameters of macro-scale models"* from the Conclusion section, and used words like *"tentatively"* and *"speculate"* in the m/s to highlight the uncertain nature of some statements (P21L6 and P22L16). We also added "*we argue that*" to a statement in the Abstract to emphasize that it represents our opinion (P1L11).

The revised m/s provides a table with qualitative interpretations of intervals of the performance metrics (see Table 4 of the revised m/s). The text has been changed accordingly.

We felt that for the present m/s a very strict separation between the Results and Discussion section would mean we would have to address the seven research questions derived from the main objective in the Results section and again in the Discussion section, which would deteriorate readability (since the reader would have to go back and forth between the Results and Discussion sections for each question). By making the distinction between discussion and conclusions more clear in the text itself we have removed any doubt whether we speculate or draw a conclusion.

I see a great value in confronting model simulations with independent observations to assess model skill and credibility, and in particular to identify model weaknesses and room for improvement. Therefore I think that the analysis presented here has potential to become a valuable addition to the scientific literature, after properly considering the reviewer comments

and some additional points of my own, as explained above and below. Since this requires a major revision of the manuscript, I may have to send the revised paper out for another round of reviews.

We thank the editor for the positive and constructive comments.

Below, I listed some of my own thoughts about your responses to Reviewer #1 and #2, followed by a list of comments I had when re-reading your manuscript in the light of the reviewer comments. Please submit, along with the revised manuscript, point-by-point responses, linked to a list of changes in the revised manuscript. The latter can also be a tracked-changes version of the revised manuscript. Thank you for your contribution to HESS and for your willingness to produce the best possible outcome for the scientific community.

Re: REVIEWER #1

Specific Comment 1: I found a range of other statements of opinion that were not clearly separated from facts. I hope you will clear out those, too.

See response before. We have more clearly separated opinions from facts in revising the m/s.

Specific Comment 2: I appreciate the authors' willingness to conduct additional analysis and look forward to seeing the results. I would propose to also add a discussion in the paper of the expectation that the data should scatter around the Budyko curve and present the data points used in the present study within the Budyko graphs to support this claim. This may also reveal any bias in precipitation or net radiation forcing, as dots outside of the Budyko envelope indicate violation of the mass or energy balance, as long as changes in storage are excluded (therefore long-term averages!).

Thank you for the useful comment. We have added Budyko density plots for three models without potential evaporation data but with net radiation data (HTESSEL, JULES, and SURFEX; see Figure 3). However, to add these models we had to exclude northern regions (>50°N/S) from the analysis since the majority of the net radiation is converted to sensible heat in these northern regions (see Kleidon et al., 2014). We have changed the m/s text accordingly (see P14L8-18). In addition, we have removed the statement that the models should scatter around the Budyko curve, since we agree with the reviewer that this does not necessarily have to be the case, given the empirical nature of the Budyko curve. However, we still plot the Budyko curve in the density plots, but explicitly mention in the text that the curve merely serves as visual reference and should not be used to judge the quality of the models.

We also produced a Budyko plot using the observations. However, after much consideration we opted not to include it in the m/s because it would require us to produce many of these plots given that each model employs a different method to compute the available energy for evaporation. Although these plots would certainly reveal if a particular catchment exceeds the

water limit and thus underestimates precipitation, this information is already available in Figure 1.

Kleidon, A., Renner, M., and Porada, P.: Estimates of the climatological land surface energy and water balance derived from maximum convective power, Hydrol. Earth Syst. Sci., 18, 2201-2218, doi:10.5194/hess-18-2201-2014, 2014.

P5L10: Please add the explanation to the paper and also explain what the different performance metrics mean in a practical sense. See also my comment below, wrt. P6L30.

We assume the Editor refers to P6L10 rather than P5L10. We refer to Table 3 for the calculation and meaning of the runoff signatures.

P6L10: I did not realise that sigma values were calculated based on a gridded data set, rather than the catchment data. I do not understand your rationale for this. Why use an observation-based model as reference here, if you claim in the introduction that the models were evaluated using observations from 966 catchments as reference? Please modify your analysis for consistency or add a compelling explanation to the text. The explanation given in your response seems to undermine your original rationale for using the catchment data.

We appreciate the comment. The sigma values are constant among the catchments and are merely used to make the values of the different signatures more comparable (i.e., to "normalize" the values of the signatures). The relative differences in scores among the models are still completely determined by the observations. We use the observation-based GSCD rather than the observations for determining the sigma values because the GSCD provides a more globally representative picture of the spatial variability of the signatures than an unevenly distributed set of observations would. We could have used the observations to determine the sigma values but this probably would not have changed the results.

We realize that our explanation of the sigma values was not sufficiently clear in the original m/s and we have therefore improved the explanation (see P6L23-28).

Tables 4 & 5: I do appreciate the detailed information, but I also agree that it is hard to read and interpret. Instead of the apparently continous colour scale, I would suggest three colours consistent with what you consider "unsatisfactory", "satisfactory" and "good" performance, or something along these lines. If this is more illustrative, you could also consider moving the table into the SI and provide bar charts instead, with a bar for each model and the different performance measures on the horizontal axis. Just an idea, not sure if it would help.

Thank you. We have changed the colors of the tables to reflect the qualitative interpretations (i.e., the continuous color scale has been replaced with a discrete one; see Table 5). We feel that in this way the tables are sufficiently clear, eliminating the need for bar plots.

Re: REVIEWER #2

General comment: Please add a discussion of the reviewer's concerns about what is considered "poor" or "satisfactory" performance to the paper and justify your own criteria up front, before discussion the results. Please also make very clear what are the main insights to be gained from your analysis, also in the context of your 2016 WRR paper.

We have added qualitative descriptions for values of the different performance metrics (see Table 4), and added some text to put the results of Beck et al. (2016) in the context of the present results (see P18L13-14).

Specific comments:

1) I think the reviewer made a very good suggestion here, and I do not see a reason to stick to the eartH2Observe collection, if additional insights could be gained by adding one more model realisation. However, if this addition would not add new insights or change the conclusions, you may as well just discuss this issue in the paper and not "contaminate" your analysis with an additional simulation data set.

We appreciate the suggestion, but Table 7 of Beck et al. (2016) shows that HBV with spatially-uniform parameters performs overall worse than two models but better than seven models. Thus, while HBV with spatially-uniform parameters performs indeed quite well among the models, it certainly did not perform beyond the range of other models. Accordingly, the fact that HBV-SIMREG outperforms the other models is really mainly attributable to the calibration and regionalization, and our conclusions and insights would not change if we were to include HBV with spatially-uniform parameters in the current analysis. We now discuss this in the revised m/s (P18L13-14): *"In their study, Beck et al. (2016) show that HBV using spatially-uniform parameters performs within the range of the other models, confirming that the relatively good performance of HBV-SIMREG stems from the regionalization exercise."*

2) Please include a discussion of the reviewer's points in the context of your previous paper in the current manuscript. Please also refer to my comment below wrt. P17L22-, and add a table as suggested by the reviewer, or point to the exact table in Dutra et al. (2015), if such a table exists. I have not found it. You may put this table in the SI, if you feel that it would disrupt the flow of the paper, but please discuss it in the manuscript. Your response about the scope of the models wrt. representing reality at scales <5000 km2 and implications for the verifiability of the models is also worth including in the discussion, in my opinion.

Text was added to the revised m/s to put the results of Beck et al. (2016) in the context of the present results (see the preceding comment). Furthermore, we now explicitly point to Table 4.1 of Dutra et al (2015) in the Introduction and Simulated runoff sections of the revised m/s, and in the Caveats section of the Methodology we now mention that some of the GHMs have been explicitly designed to estimate runoff in small catchments (lines P9L8-12): *"... some of the*

*models (notably the LSMs) were not traditionally developed to estimate daily runoff for such small catchments. Some of the GHMs, on the other hand, have runoff estimation in small catchments among their primary aims (e.g., LISFLOOD, WaterGAP3, W3RA, and HBV-SIMREG), and four GHMs were even explicitly calibrated against observations (LISFLOOD, SWBM, WaterGAP3, and HBV-SIMREG; see Section 4.4 for specifics)."*

3) Please add NSE values as suggested and a discussion of their meaning to the manuscript. The current blunt dismissal of NSE in the manuscript is not very helpful.

We have added NSE scores to the Supplementary information and added the following text to the revised m/s (P10L33-P11L7): *"Although the NSE has been widely criticized for being overly sensitive to the magnitude and timing of peak flows (e.g., Schaefli and Gupta, 2007; Jain and Sudheer, 2008; Criss and Winston, 2008; Gupta et al., 2009), we did calculate NSE scores to allow the present results to be put in the context of previous continental- and global-scale studies (see Supplementary material Table S1). For most models slightly negative median NSE scores were obtained, similar to Zhang et al. (2016), who evaluated the monthly and annual runoff estimates from 14 (uncalibrated) macro-scale models in 644 large Australian catchments (>2000 km$^2$). Our scores are, however, slightly lower than those obtained by Lohmann et al. (2004) and Xia et al. (2012), who evaluated the daily runoff estimates from four (uncalibrated) macro-scale models in about a thousand small-to-medium sized USA catchments (<10000 km$^2$), but this is probably attributable to the high quality of the USA forcing data. They are also somewhat lower than those obtained by Decharme and Douville (2007), who evaluated two (uncalibrated) models in 80 large catchments (>100000 km$^2$) around the globe, but this can be explained by the much larger catchment sizes."*

P6L14: If I am not mistaken, the reviewer is worried that the mean difference may obscure differences in timing, meaning that a value of 0 may be considered a very good result, whereas in reality over-estimation in one period is compensated by under-estimation in another. If adding Stdev is not justified because of deviations from normality, please at least discuss the meaning of the D values and explain how they should be interpreted in the context of other metrics. You could also think of providing some combined index that combines different metrics to produce a model-data correspondence between 0 and 1.

The OS metric (Equation 4) can be considered a "combined index" since it combines the performance in terms of signatures and temporal variability. We have added the following text regarding D: *"It should be noted that, although D provides a valuable estimate of the overall performance, a good D value may reflect an overestimation in one region that is compensated by an underestimation in another region."*

P17L34: Does this mean that the procedure was not sufficiently documented? Please explain in the text.

We are not sure which procedure is referred to here. P17L34 refers to the fact that the parameters of macro-scale models in general tend to be based on "expert opinion".

EDITOR's COMMENTS:

In addition to the reviewers' points, I would like to see clear statements about the expected model skills and the associated performance metrics and what would be considered satisfactory model performance. Just to grab an example, what do values of spatial correlation in the runoff coefficient between 0.3 and 0.67 mean and what values would be considered satisfactory? If the expected model skill is to predict trends in mean annual runoff, then neither of the models discussed in this paper appears to be useful, as they do not reproduce the trends. In this context, the mention of "studies assessing the hydrological impacts of climate change" in the abstract warrants a discussion of the suitability of these models for this purpose.

Thank you. We have added qualitative descriptions for values of the different performance metrics (see Table 4 of the revised m/s).

P2L17: Really all studies?

We have added "to our knowledge" and added three additional studies.

P2L24: In Table 1, most studies include the daily time scale, whereas in your study, you also included monthly runoff data in the evaluation. Could you clarify what you mean and give examples for the additional insights gained. This could then be moved to the discussion.

In the Introduction we have added a sentence that using only monthly data precludes analysis of the shape of individual flow events (P2L26).

P2L28: What are the new insights gained by including more models?

Each hydrological model behaves in a unique way. By using more models we sample a larger part of the behavioral space, which may lead to more generalizable insights relevant to a wider audience.

P3L5: Need to discuss what is meant by "well". Reproduction of a time series?

Essentially, we want to know how well the different models mimic reality, whereby the observations are assumed to reflect reality.

P3L13: Really daily? On P4L25 you state that daily and monthly observed data was used, and on P6L2 you clarify that 325 catchments (i.e. 30%) had only monthly data.

Thank you for bringing this to our attention. This should have been both daily and monthly. We have corrected the m/s.

P3L14: More reliabale conclusions than previous studies? In how far?

Correct, more reliable compared to previous studies. We are not sure what is meant by the second part.

P15L9: Could you provide a clearer analysis illustrating the similarity in the produced trends? Fig. S1.8 contains the data, but it is hard to tell in how far the colours are similar between models for the same catchments.

The main message we wanted to convey is that the results are very similar among models and as the Editor points out the figure conveys that. Although we appreciate the suggestion to add a figure showing the standard deviation among models for each station, we are not sure if this is of interest to a wide enough audience, and the m/s is already quite lengthy.

P6L17: Do you mean spatial variability or variance? How was it calculated?

We appreciate the comment. We mean the spatial variability as expressed by the standard deviation (not the variance) calculated from the GSCD signature values of the catchments. In the revised m/s we have improved the explanation of the sigma values (see P6L23-28).

P6L30: A value of 0 for Pearson's r means no relationship, negative values imply a negative relationship. Clearly, one would not say that 0 is better than -0.2, so your statement here needs to be more specific. Please specify what values would be considered satisfactory.

We have added qualitative interpretations of the correlation classes (see Table 4 of the revised m/s). Based on this classification, correlations of 0 and -0.2 are both considered to be "poor".

P12L34: Was the performance satisfactory in other regions? How do you define satisfactory performance?

Please see Table 4 of the revised m/s.

P17L4-7: Calibration does not compensate for lack of process understanding or other model deficiencies, it just reduces their effect for data similar to the calibration data. This is precisely what your results reflect. The more similar the calibration data and the metrics used in the calibration to your validation data, the better the perceived model performance. What can you tell about the degradation of the model performance under climate change? I would argue that a well-calibrated model is more likely to create a false sense of accuracy in prediction mode than an uncalibrated model.

Thank you for your comment, we partially agree in that good performance in the past does not guarantee good performance under future conditions. On the other hand, we would argue poor performance in the past all but guarantees poor performance in future. We do accept that (over-)calibration can give rise to overconfidence, however, and have replaced "compensate" with "account" (see P17L13).

P17L22-: Earlier in the manuscript, you refer to this section for details about the calibrations performed. The details given here are insufficient, as they do not contain any information about what kind of parameters were calibrated. Were these only parameters relating to physical catchment properties, or also related to e.g. vegetation properties, land cover etc.? What were the objective functions and how exactly do the objective functions and calibration data sets overlap with those used in the performance analysis presented here? Perhaps a table would be helpful here.

For WaterGAP3, SWBM, and HBV-SIMREG more details can be found in the provided references (the reference for WaterGAP3 has been added in the revised m/s). It is difficult to pinpoint for each parameter which process or physical property it is related to, since many of the parameters in the GHMs and LSMs are conceptual in nature and integrate several processes within a catchment (e.g., the BETA parameter of LISFLOOD and HBV-SIMREG).

The original m/s described the objective functions used for calibrating each model (P17L29-P18L17 in the revised m/s).

P18L29-: What evidence can be presented to claim that WFDEI precipitation is more biased than PRISM precipitation? In order to support your claim that bias in MAR is due to bias in WFDEI precipitation, you could use PRISM precipitation as input and see if the bias goes away. Otherwise, this seems like speculation.

We used PRISM as reference because, compared to PRISM, WFDEI incorporates considerably less gauges and less sophisticated orographic corrections. The following text has been added to the revised m/s: *"It is conceivable that biases are present in the WFDEI P data, because: (i) the monthly CRU dataset, which has been used to correct the WFDEI dataset, is based on only a subset of the available gauges and does not explicitly account for orographic effects; (ii) in sparsely gauged regions the correction using CRU is more likely to deteriorate rather than improve the estimates; and (iii) the Adam and Lettenmaier (2003) gauge undercatch correction factors are based on interpolation of a very sparse sample of gauges and thus subject to considerable uncertainty."*

Besides the comparison between PRISM and WFDEI, another strong line of evidence indicating that WFDEI contains biases is the consistent bias patterns among the models (e.g., all models generally underestimate flows in mountainous regions).

PRISM data are only available at a monthly resolution which is insufficient to drive any of the models considered in our study.

P18L34: Please provide a quantitative analysis supporting "comparable bias pattern". At first eye shot, Panels a and b may look similar, but this is hardly a defendable analysis and the conclusion that the P bias propagates into MAR bias is hence not clearly supported by evidence.

Thank you for the comment. The correlation between the precipitation and MAR bias values is 0.58, suggesting there is a moderately strong relationship (correlations between 0.4 and 0.6 are considered "moderate", see Table 4 of the revised m/s). We mention the correlation coefficient in the revised m/s (P19L14).

P20L24: What is a "multi-parameterization ensemble"?

This is explained in the preceding sentence (P20L21-23 in the original m/s): *"HBV-SIMREG differs from the other models because it represents a so-called 'multi-parameterization ensemble', which means the model was run multiple (ten) times globally using different (regionalized) parameter sets representing different catchment response behaviors (Beck et al., 2016)."*

Conclusions:
4.: Clearly, if a model is calibrated to maximise the performance metrics ("performance metrics incorporated in the respective objective functions"), it will "perform" better than if it is not calibrated or calibrated to maximise some other metrics. Therefore, this line of argument is circular and the resulting recommendations (statements starting with "should be") potentially flawed. I consider both statements starting with "should be" as controversial and clearly the authors' opinion, not necessarily a result of the data presented. Please re-word or put your opinions in a separate paragraph, clearly identifiable as opinions.

Thank you, agreed. We removed both statements containing "should" from conclusion (4). Although we appreciate the suggestion, we did not change conclusion (5) because the uncertain nature of this conclusion is already highlighted by using the words "we speculate".

5.: The statement in this form is not supported by the evidence presented. The evidence merely shows that the WFDEI P deviates from PRISM P, apparently in similar locations where simulated MAR also deviates from the observed. See my comments above, with respect to P18.

We refer to our response to P18L29.

[revised manuscript text omitted]

---

## Author Response (AR2)

Dear Dr. Schymanski,

We hereby submit the revised version of our m/s entitled "Global evaluation of runoff from ten state-of-the-art hydrological models". Our responses to each of your and the reviewer's comments are presented in green font below. The main changes in this revision are:

1) improved the colormap of Table 5 by maximizing the tone, hue, and saturation differences between the performance classes, to improve the readability;
2) produced Budyko plots based on all grid cells, instead of only grid cells at latitudes <50°;
3) computed normalization values for the runoff signatures based on the observations rather than the Global Streamflow Characteristics Dataset (GSCD), to reduce the complexity of the analysis; and
4) introduced some textual improvements and clarifications.

We want to sincerely thank you for handling the m/s and trust that this revision is suitable for publication.

Sincerely,
Hylke Beck (on behalf of all co-authors)

Comments to the Author:
Dear authors,

I agree with the reviewers that the manuscript has improved substantially, but I think that some of the remaining points are not just minor, as they could potentially change some of the conclusions.

In Report #1, the reviewer doubts that the conclusion about the potential of multi-parameterization ensembles is backed by the analysis and asks for additional analysis to support the statements. The report further highlights a potential flaw in the evaluation of SWBM runoff results and the potential sensitivity of the conclusions to uncertainty in the daily climate data. If it is not feasible to remove these ambiguities, they should at least be clearly discussed. In Report #2, the reviewer echoes my original concern about the normalization of the performance metrics using the GSCD data. In contrast to your assessment that a different choice of normalization would probably not have affected the results, the reviewer suggests that the normalization may affect the conclusions. I think this is too important to be left to speculations either way and I would like to ask you to provide evidence (in the manuscript) that the normalization does indeed not affect the results and conclusions. As pointed out by the reviewer, Table A5 is still very difficult to comprehend. I am not sure if a heat map or bar chart would help, but perhaps a reduction of the table to convey the main points needed to support the conclusions would be helpful, while the full table could be given in the SI. I also concur with the reviewer that the rationale for limiting the Budyko plots to the lower latitudes (<50 degr.) is not clear. Why not include all data? You should at least show the same plots including all data in the SI and briefly discuss any striking differences.

We have improved the colormap and increased the font weight of Table 5, which makes the table easier interpret. We have considered presenting the results in bar plots, but feel this would not not necessarily be an improvement due to the large number of bar plots it would require. We are not in favor of making the results less accessible by moving them to the Supplementary information.

We now normalize the runoff signature values using the observations rather than the GSCD data, as requested, which resulted in very minor changes in the scores in Table 5 and no changes to the conclusions.

We have improved the Budyko plots by using all data, instead of only data for latitudes <50°.

In addition to the reviewer comments, I still have a few comments of my own:

EC1 - The paper contains a number of value statements that are not clearly backed by the analysis, and some could even be misleading. In several occasions, you use the word "reliable", which implies some degree of trust and robustness. When referring to models, I would propose to replace "reliable models" by "accurate model simulations", as your assessments are drawn

from correspondence between particular observations and model simulations, which strongly depend on the calibration procedure and data. When referring to conlusions, I really see no evidence to suggest that your conclusions are more reliable and generalizable than those presented in other papers, so I would remove this statement and instead be more specific about the additional insights gained.

We have replaced "reliable" by "accurate" throughout the m/s, as suggested. In addition, we have removed the statement that our results are more reliable and generalizable than previous studies from the Introduction.

EC2 - I also think that the value of calibration is over-stated and not discussed in a balanced way. As I alluded to before, calibration does not necessarily account for, nor compensate for inadequate understanding or missing information. All it evidently does is to reduce mismatch between model simulations and (calibration) data. Therefore, I would propose to replace the sentence on P17L12, "Calibration is a prerequisite for both conceptual and physically-based hydrological models to provide reliable runoff estimates, to account for (i) the impossibility of measuring all required model parameters at the model application scale, (ii) lack of process understanding..." by something like:
"Calibration is necessary whenever model parameters cannot be estimated a priori and is also used in both conceptual and physically-based hydrological models to reduce simulation-observation mismatch, commonly assumed to be caused by (i) lack of process understanding..."
Your result that calibration against one signature does not improve the model-observation mismatch in another signature does not fill me with confidence that the calibration makes the model any more "reliable", but you are of course free to express your own opinion, as long as you mark it as such.

Thank you for the comment. In an effort to soften the sentence we have replaced "prerequisite" with "important" and added "more" before "accurate" ("reliable" in the previous revision). We certainly agree that calibration does not fully compensate for inadequate understanding or missing information, but in our opinion it does help to some degree (hence our use of "more accurate"). We want to thank you for proposing a sentence, this is really appreciated. However, we have two issues with the sentence. First, it does not refer to runoff which is the topic of this paper. Second, it implies that for some models their runoff-related parameters can be estimated *a priori* and therefore do not need calibration. Such models do not exist, because (i) there is simply no way to obtain accurate information on important subsurface hydrologic characteristics that control runoff generation processes at the model application scale (see, e.g., Beven, 1989; Blöschl and Sivapalan, 1995), and (ii) all hydrological models (even high-resolution physically-based models) represent gross simplifications of reality and therefore even if you would be able to estimate all parameters *a priori* they would still non behave realistically. Perhaps for other hydrologic variables which are not the topic of this paper (e.g., lake outflow) it might be possible to estimate the relevant parameters *a priori*, but for runoff this is simply not possible.

Beven, K. J.: Changing ideas in hydrology — the case of physically-based models, Journal of Hydrology, 105, 157–172, 1989.

Blöschl, G. and Sivapalan, M.: Scale issues in hydrological modelling: A review, Hydrological Processes, 9, 251–290, 1995.

EC3 - In any case, the reader is entitled to know what the calibration actually entailed, i.e. how much overlap between calibration and validation data there was for the different models and what process parameters were actually calibrated, be it conceptual or physically-based.

We certainly agree, and therefore outline the calibration exercises performed for each model on page 16 line 29 to page 16 line 17. For more details we refer to the corresponding publications.

EC4 - This is crucial for the missing discussion of the reliability of the models or multi-model ensembles in assessing climate change impacts, which was also requested by Reviewer #2 in Report #1. For example, if one calibrates parameters relating to vegetation water use, one should expect that the calibration will become inadequate as vegetation properties change, e.g. due to change in rainfall or atmospheric $CO_2$ concentrations. Please add a balanced discussion about the utility of this approach for climate change analysis or remove reference to climate change analysis and specify more clearly what the intended use of the models and multi-model ensembles is. I doubt that the intended use is to "mimic reality", where "reality" is expressed by a set of observations. One could use a purely statistical model for this, or, to be provocative, a set of polynomials of an arbitrary level of complexity.

It is true that the calibrated parameters becomes less representative when the model is subjected to new conditions, which we now explicitly mention in the m/s ("for climate projections one should bear in mind that calibrated parameters become less valid when the model is subjected to climatic conditions it has never seen before (Knutti, 2008)"). We have also added to the Introduction that the objective of the eartH2Observe project is "to develop a global reanalysis of water resources that supports efficient water management and decision making", to avoid any confusion about the intended use of the models.

I am sorry to ask for additional work at this stage, but given the potential impact of the concerns expressed in this round of review, I cannot accept the paper for publication in HESS in its current form. I hope that this last revision will result in a valuable paper.

We understand and appreciate your time and constructive help to further improve the m/s.

REFEREE #1

OVERALL RATING:

I do appreciate the effort of Beck et al to further improve their manuscript, with respect to both textual clarifications as well as additional analysis in the re-arranged Budyko space. Nevertheless I do have several specific comments and suggestions as listed below:

We want thank Dr. Gudmundsson for his insightful comments.

SPECIFIC COMMENTS (SC):

**
** SC1:
**

I am still not convinced of the authors strategy to normalise their performance metrics using the GSCD data. While I do understand the argument that this data set does only serve normalization purposes and may provide more consistent global estimates of the standard deviations, I do still believe that utilizing this data set makes the analysis more complicated without a significant benefit. Nevertheless, I must also admit that the choice of the data set used for normalization (either GSCD or station data) will most likely impact the overall conclusions drawn by the authors.

We now use the standard deviation of the observed values to normalize the signatures, resulting in very minor changes in the scores in Table 5 and no changes to the overall conclusions. We have modified the text in the Methods section accordingly.

**
** SC2:
**

Overall the overwhelming information content makes Table 5 very difficult to read. I had to spend several minutes staring at the numbers before I could get any meaningful information form it. Unfortunately the colours for "fair", "moderate" and "good" are almost indistinguishable on my screen, which does render this information almost meaningless. Consequently I would like to encourage the authors to replace this table with one or two figures (e.g. bar charts, heat maps) and to present the detailed quantitative information in the supplement.

Thank you for the comment. We now use a colorscale that maximizes the difference in tone, hue, and saturation between the different performance intervals (produced using ColorBrewer, see http://colorbrewer2.org), to improve the readability of the table. We are now confident that the table is easy to interpret.

**
** SC3:
**

If I got it correctly: Table 5 presents global means whereas Table S1 shows global medians. This is inconsistent and should be adjusted accordingly.

We would indeed have prefered to use means for the NSE values for consistency. However, the use of means would render Table S1 meaningless because the values would be completely dominated by some catchments with extremely low NSE values. We therefore decided not to use means, although we appreciate the suggestion.

**

** SC4:

**

I do really appreciate the authors effort to present "budyko-plots" for a wider range of options. I am, however, not convinced by their choice to mask out regions >50deg N/S. While I can follow the argument that most of Rn will feed into the sensible heat flux, I do not understand why this would be an argument to exclude this from the analysis. To me this reads like an attempt to hide/mask out unwanted information. Note also that much of the Original work if Budyko was based on Rn AND included also information form high latitudes.

We have redone the plots to include all grid cells and changed the text accordingly. Thank you for the suggestion.

**

** SC5:

**

In some sections there is still a tendency of value statements and a fair degree of "NSE-bashing" which I think is not necessary. I fully support the authors choice not to focus on NSE I think that a more neutral wording would be valuable for the scientific discourse

We have removed "widely" from the sentence "the NSE has been widely criticized for being overly sensitive to the magnitude and timing of peak flows" to soften our critique.

REFEREE #2

The authors have to a large extent responded to the comments of the reviewers and the editor. After reading the authors` response as well as the revised version of the manuscript, a few important suggestions remain that I would like to be addressed in the final publication.

We would like to thank the reviewer for their comments and suggestions.

I think that my concern regarding the conclusion about the potential of multi-parameterization ensembles (no. 2 of my review) not being backed by the analysis was not clarified.
*"2) The study of Beck et al. (2016) also indicates that performance of the HBV-SIMREG model results that are not derived as the ensemble mean of 10 runs with 10 different parameter sets but just 1 (derived from the most similar donor catchment) perform only slightly worse than the*

*ensemble mean and better than the other models (Tables 6 and 7 of Beck et al.). Therefore, the conclusion that the fact that HBV-SIMREG with 10 runs performs better than the ensemble mean of all models tentatively suggests that a multi-parameterization ensemble for a single, sufficiently flexible model could replace multi-model ensemble studies (p. 20), is not backed by the analysis in the manuscript. I suggest including the HBV-SIMREG variant with 1 run/parameter set only, and consider the result when formulating such a conclusion. In addition, it should be taken into account (and explained very clearly in the manuscript) that HBV-SIMREG only computes runoff in 0.5° grid cells and not river discharge, as grid to grid lateral routing including the impact of lakes and wetlands as well as water abstraction are not simulated by this model."*

The HBV-SIMREG output that is not the ensemble mean of 10 runs with 10 different parameter sets (evaluated in the manuscript) but, like in the case of all other evaluated models just the output derived from using one (calibrated) parameter set does need to be included in a study that concludes (P21L8ff, also P23) that " a multi-parameterization ensemble for a single, sufficiently flexible mode provides performance comparable to a multi-model ensemble". Please either add the performance indicators of such a values HBV-SIMREG variant either to Table 5, the supplement or just provide selected values in the text. In addition, the conclusions may need to be extended by a consideration regarding the benefit of multi-model ensembles (with different model structures) for e.g. climate change impact studies.

We would like to clarify that our study does not *conclude* that "a multi-parameterization ensemble for a single, sufficiently flexible model provides performance comparable to a multi-model ensemble". Instead, our results "*tentatively suggest* that a multi-parameterization ensemble for a single, sufficiently flexible model provides performance comparable to a multi-model ensemble". The words "tentatively" and "suggests" highlight the uncertain nature of this finding. In addition, immediately after this statement we further emphasize the inconclusive nature of this finding with the words "If this is confirmed, ...".

Moreover, Table 7 of Beck et al. (2016) shows that HBV with spatially-uniform parameters performs, on average, worse than two models but better than seven models, meaning that while HBV with spatially-uniform parameters performs indeed fairly well among the models, it certainly did not perform outside the range of the other models. Thus, the fact that HBV-SIMREG outperforms the other models is therefore mainly attributable to the calibration and regionalization efforts, and our conclusions and insights would not change by including HBV with spatially-uniform parameters in the current analysis. We explicitly mention this in the m/s: *"In their study, Beck et al. (2016) show that HBV using spatially-uniform parameters performs within the range of the other models, confirming that the relatively good performance of HBV-SIMREG stems from the regionalization exercise."*

Furthermore, the current study is about evaluating the eartH2Observe collection of models, and HBV with spatially-uniform parameters is not part of this collection.

We agree that a multi-parameterization model may not perform as well when it is subjected to climate projections, which is one of the reasons why we push for additional research on this topic.

B
Looking at Dutra (2015), I noticed that it is not clear in the manuscript, what output variable of the models have actually been evaluated by comparing their value to the streamflow at the outlet of small catchments. On P4L10, the variable is only described as "simulated daily (non-routed) runoff (mm/d)." I think that for a proper comparison to observed streamflow, the "simulated runoff" should be the amount of water that is simulated to enter the stream each day, i.e. is the storage of water in the groundwater and the surface water bodies (if they exist in the models) taken into account. Could you please clarify this the text.

However, Dutra (2015), section 4.8, writes about SWBM as used in Tier1, i.e. for the study described in the manuscript, that SWBM runoff results should not be evaluated at the daily time scale because the model in the Tier1 version does not include the impact of groundwater storage on streamflow:

"The OS13 model version incorporates a delayed formation of streamflow to account for the (sub-surface) transport of runoff water to the stream-gauge site, whereby the runoff water is stored in a groundwater storage before it is transformed into streamflow. Even if this mechanism is crucial to estimate daily streamflow dynamics we excluded it from the model because (i) this reduces the computational effort considerably through the removal of a parameter and because delayed streamflow is not calculated and (ii) it does not impact soil moisture and ET. All results related to runoff in this study are only valid on the monthly time scale where runoff and streamflow are rather similar."

However, in the ms, daily SWBM runoff seems to have been used.

Thank you for the comment. For all models we used simulated specific runoff depths rather than routed discharge volumes, because the latter was available for only a subset of models. These runoff amounts include both surface and subsurface runoff and these thus include groundwater contributions. We now explicitly mention in the text that "for each model the sum of the subsurface and surface runoff components was calculated" (page 4 line 22), to avoid any confusion. We agree that it would be better to compare simulated discharge to discharge observations but note that the simulated runoff is similar to the simulated discharge since we focus catchments with an area comparable to the model's grid.

We want to note that Rene Orth, the developer of SWBM, is co-author of this study, and thus supports the daily evaluation performed in this study. We are also aware that SWBM does not include a baseflow component and therefore performs worse for baseflow-related metrics, which we explicitly mention in the text (page 14 line 24).

C
Section 3.3. Caveats

I suggest adding one further caveat which is according to my experience very important: That the results of the model evaluation may strongly depend on the applied climate data set, in particular because the evaluation considers small time periods (days) and areas (0.5° grid cells) for which e.g. the uncertainty of precipitation data is very high.

We appreciate the suggestion and have added the forcing data as an additional caveat ("the forcing data quality has an important influence on the evaluation results that should not be overlooked", page 9 lines 15-16).

Minor
P10: Replace reference Döll and Flörke (2005) by
Döll, P., Fiedler, K. (2008): Global-scale modeling of groundwater recharge. Hydrol. Earth Syst. Sci., 12, 863-885.

Changed, thanks for the suggestion.

P17L7: Replace: "assess the impacts of climate change" by "assess the impacts of past climate change"

Changed as suggested.

[revised manuscript text omitted]